



# Reconstructing Nineteenth-Century River Water Levels with Transformer-Based Computer Vision

Malte Rehbein[1,2]

[1]Chair of Computational Humanities, University of Passau, Germany
[2]Max Planck Institute of Geoanthropology, Jena, Germany

**Correspondence:** Malte Rehbein (malte.rehbein@uni-passau.de)

**Abstract.** We convert nineteenth-century Bavarian Danube gauge charts (1826–1894) into daily water-level series referenced to gauge zero through a novel semi-automated workflow combining light document pre-processing, dewarping, transformer-based line extraction, pixel-to-curve calibration, and targeted human checks. A curated ground-truth sample supported benchmarking and uncertainty quantification. Across three representative gauges (Neu-Ulm, Vilshofen, Passau), the pipeline attains high series-level accuracy (mean composite score 0.979) while reducing manual effort by roughly an order of magnitude relative to full manual digitisation. Outputs include versioned datasets with page-level provenance, confidence scores, and methodological descriptors to ensure transparency and reuse. The approach offers a replicable template for rescuing analogue hydrometric records and enabling long-term analyses of extremes, regulation impacts, and ecological context. Data are openly available under CC BY 4.0 (Rehbein (2025); DOI: 10.5281/zenodo.17296750).

## 1 Introduction

Long hydrological series allowing among others flood frequency analysis are essential for putting contemporary extremes, regulation impacts, and ecological responses into context (Brázdil et al., 2006; Neppel et al., 2010; Lucas et al., 2024; Wetter, 2017). Yet large volumes of nineteenth-century hydrometric information remain locked in historical archives, limiting their contribution to present analyses. This paper addresses that gap for the Bavarian Danube by converting hand-drawn annual gauge charts for the years 1826–1894 into daily, machine-readable water-level (stage) series with transparent uncertainties and page-level provenance.

Our sources are printed quarterly templates with a regular day–level grid and handwritten hydrographs (see Fig. 1) for an example). They exhibit typical historical heterogeneity: paper warp and book curvature, stroke shadowing and show-through, inconsistent drawing styles, unit transitions, and occasional changes to gauge zero. Any practical reconstruction must therefore be robust to variable image quality while keeping human effort modest and the process auditable.

We make the following contributions. **(i)** A pragmatic, semi-automated workflow (HWLR) that combines light, grid-aware pre-processing, optional dewarping, transformer-based line extraction (using *LineFormer*), and a pixel-to-curve calibration tied to documented gauge zero, with targeted human checks at clearly defined checkpoints. **(ii)** A curated ground-truth sample and an evaluation protocol that report standard errors (RMSE/MAE, Pearson's $r$) alongside a peak-aware composite score that


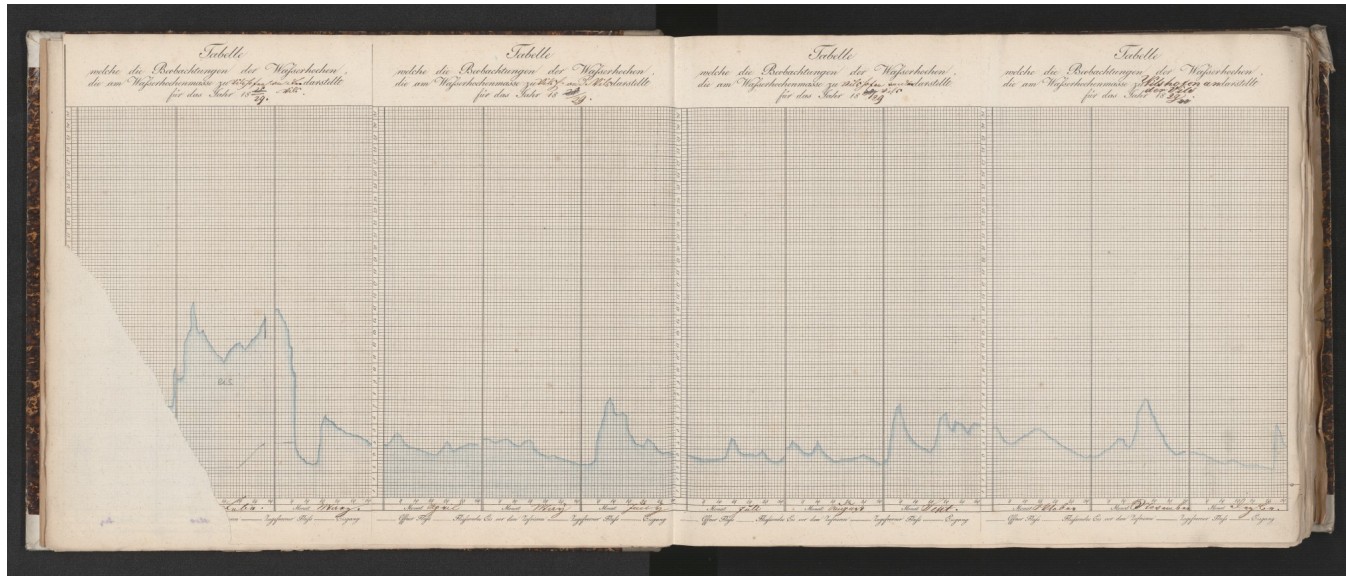

**Figure 1.** Sample from the Vilshofen gauge chart (year 1829). It also illustrates one of the (rare) cases of paper blight. Source: BayHStA BLW 29, page 7.

better reflects the historical charts' intended use. **(iii)** A case study on three representative gauges (Neu-Ulm, Vilshofen, Passau) demonstrating high series-level fidelity while substantially reducing manual digitisation effort. **(iv)** An open, versioned release of the reconstructed data with month-level provenance, uncertainty flags, and code pointers to enable verification and reuse (CC BY 4.0; DOI: 10.5281/zenodo.17296750).

The remainder of the paper introduces the sources and their structure; details the reconstruction method and implementation
choices; presents accuracy and efficiency results, including an analysis of systematic errors and mitigations; and discusses limitations, generalisability, and maintenance as a living dataset. Together, these elements offer a replicable template for mobilising historical hydrometric charts at scale.

## 2 Description of Extant Records (Materials)

### 2.1 Historical Context

The systematic use of hydrometry in Western Europe was largely forgotten until the Renaissance, however. It was not until the 18th and 19th centuries that sustained, systematic approaches to river monitoring re-emerged, driven by growing urban populations, expanding infrastructure, and repeated flood disasters. By the early 1800s, a network of river gauges and water level charts had been established across Europe, including in the Kingdom of Bavaria—one of the largest German states with then 75,865 km$^2$ and access to both central European large streams: Rhine and Danube. This period also marked the rise of
large-scale land surveys and environmental data collection for purposes ranging from state administration to nation-building.




Water-level observation underpinned navigation, flood protection, and water-resources management throughout the nineteenth century. In Bavaria—mirroring broader European developments—hydrometric practice became increasingly institutionalised as river training, urban growth, and industrial land-use change accelerated. This institutionalisation is visible in administrative reforms, technical rulebooks, and the emergence of printed hydrological yearbooks and standardised gauge documentation (Bayerisches Landesamt für Umwelt, 2020); see also the historical overview provided by the Bavarian environmental administration (https://www.lfu.bayern.de/wasser/wasserstand_abfluss/entwicklung_pegelwesen/index.htm).

A concise synthesis of the hydraulic transformation of Bavarian rivers is provided by Scheurmann (1981), who traces nineteenth- and early twentieth-century interventions motivated by flood control, land reclamation, and navigation. Major rivers, including the Isar, Vils, Amper, and the Danube, were straightened, confined, and stabilised using fascines, levees, groynes, and weirs. One illustrative case is the Danube between Ingolstadt and Großmehring, where channel realignment and embankment works (ca. 1826–1830) fundamentally altered local hydraulics and floodplain connectivity (City Museum Ingolstadt: https://www.ingolstadt.de/stadtmuseum/stadtmuseum/scheuerer/museum/r-01-003.htm). Archival base maps from the Bavarian land survey (*Uraufnahme*) already display projected realignments, offering a cartographic window into early planning and design choices (Landesamt für Digitalisierung, Breitband und Vermessung: https://www.ldbv.bayern.de/mam/ldbv/dateien/faltblatt_historischekarten.pdf).

From ca. 1700 to the late nineteenth century, Danube water-level information survives in documentary registers, municipal accounts, newspapers, flood marks, and later hydrographic ledgers. Taken together, these sources enable reconstructions of flood chronologies, relative levels, and, where cross-sections or benchmarks are known, approximate stages. With the establishment of dedicated hydrographic services and printed yearbooks in the late nineteenth century, qualitative accounts increasingly dovetail with systematic instrumental observations (https://www.lfu.bayern.de/wasser/wasserstand_abfluss/entwicklung_pegelwesen/index.htm).

Administrative and legal frameworks for hydraulic engineering and hydrometric observation were codified in progressively more detailed technical guidelines. Notably, the *Technische Vorschriften für den Wasserbau an den öffentlichen Flüssen in Bayern* (21 November 1878) standardised procedures for surveying and documenting works on public rivers; programmatic summaries followed in 1888 and 1909. This lineage culminated in the *Vorschrift und Vollzugsanweisung für Flussausstattung, Flussaufnahmen und deren Verarbeitung* (1930), which remained legally binding until 1958 (Bayerisches Landesamt für Umwelt, 2020; Weigmann, 1935; Faber, 1903). These regulations unified practices for river surveying, gauge-station documentation, and the graphical representation of water levels across Bavaria.

In terms of technical terminology, German nineteenth-century hydrometric literature may refer to water levels as *hydrometrische Coten*—positive or negative water heights relative to a fixed gauge zero (*Pegelnull*)—a convention mirrored in the layout of historical charts analysed in this study (Schrader, 1884). As state hydrologic services matured, guidance converged on consistent gauge datums and reporting standards; in Bavaria, the first *Gewässerkundliches Jahrbuch* with water levels from Bavarian gauges appeared in 1895, and a unified *Pegelvorschrift* was introduced across Germany in 1935 (https://www.lfu.bayern.de/wasser/wasserstand_abfluss/entwicklung_pegelwesen/index.htm).

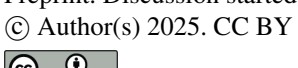

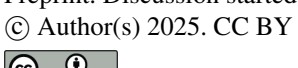

**Figure 2.** Kingdom of Bavaria in its administrative borders from 1845 (grey underlay). Danube and its tributaries (dark blue), other main rivers (light blue). HydroBasins Level 7 (red). Gauge stations for which level recordings are reported: along the Danube (dark blue) and other rivers (light blue). Gauges stations for this study (highlighted). Sources: Bayerisches Hauptstaatsarchiv, HGIS Germany, OpenStreetMap, HydroBasins, AQUASTAT (FAO).





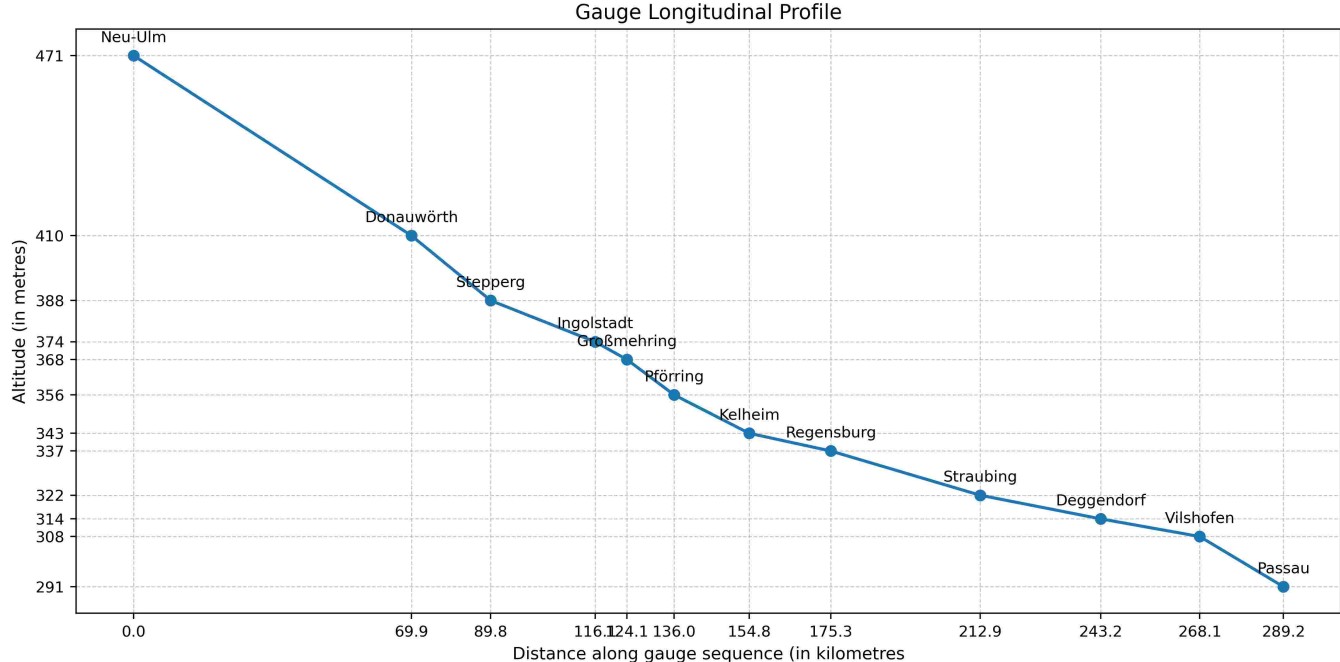

**Figure 3.** River longitudinal profile of the 19th century Bavarian Danube gauges; distances between gauge station as the crow files (standard great-circle (Haversine) distance).

## 2.2 Records

**Provenance and format.** The primary sources are archival river–gauge charts compiled by the *Bayerisches Landesamt für Wasserwirtschaft (BLW)* and today preserved in the *Bayerisches Hauptstaatsarchiv (BayHStA)* (Munich). The record extends back to the early nineteenth century and consists of a continuous, regionally extensive series of hand-drawn hydrographs on pre-printed forms. Each standard form (approx. $20 \times 34$ cm) covers *one quarter* (three months) with a daily grid, a vertical scale and unit annotations; a hand-drawn curve traces the water level at daily resolution. Four quarterly forms were glued edge-to-edge to create a single annual sheet, typically annotated with units, vertical ticks, and occasional later corrections. Series of annual sheets for a given gauge were then bound into volumes; some sheets carry additional notes on the reverse.

**Compilation practice.** Visual inspection shows that the annual sheets were not updated day-by-day but compiled retrospectively, possibly on a quarterly cadence. This implies the existence of primary daybooks or tabular ledgers from which the quarterly charts were prepared. For the Bavarian Danube, such underlying tabular sources have not been located to date; consequently, the bound charts constitute the most complete surviving record for the nineteenth century.

**Archival indexing and coverage.** The archival finding aid (*Findbuch*) maps call numbers to gauges and calendar periods, which we use to construct a station–time index for sampling and comparison. For the Bavarian Danube in the nineteenth cen-





**Table 1.** Nineteenth-century Bavarian Danube gauges in the archive; case-study gauges highlighted. Six Danube gauges from a later period are not listed here. Source: Bayerisches Hauptstaatsarchiv.

| gauge_id | gauge_name | longitude | latitude | altitude | reporting period |
|---|---|---|---|---|---|
| **DE_BY_DAN_21** | **Neu-Ulm** | **9.987** | **48.390** | **471** | **1826-1894** |
| DE_BY_DAN_22.1 | Donauwörth | 10.803 | 48.711 | 410 | 1848-1889 |
| DE_BY_DAN_22.2 | Pförring | 11.691 | 48.794 | 356 | 1848-1889 |
| DE_BY_DAN_23 | Ingolstadt | 11.428 | 48.760 | 374 | 1826-1894 |
| DE_BY_DAN_24.1 | Großmehring | 11.537 | 48.760 | 368 | 1848-1894 |
| DE_BY_DAN_24.2 | Stepperg | 11.071 | 48.741 | 388 | 1848-1894 |
| DE_BY_DAN_25 | Kelheim | 11.873 | 48.914 | 343 | 1826-1894 |
| DE_BY_DAN_26 | Regensburg | 12.102 | 49.021 | 337 | 1826-1894 |
| DE_BY_DAN_27 | Straubing | 12.574 | 48.886 | 322 | 1826-1894 |
| DE_BY_DAN_28 | Deggendorf | 12.971 | 48.809 | 314 | 1826-1894 |
| **DE_BY_DAN_29** | **Vilshofen** | **13.187** | **48.636** | **308** | **1826-1894** |
| **DE_BY_DAN_30** | **Passau** | **13.459** | **48.577** | **291** | **1826-1894** |
| DE_BY_DAN_31.1 | Passau (Ilzstadt) | 13.479 | 48.575 | 295 | 1826-1894 |
| DE_BY_DAN_31.2 | Passau (Ortspitze) | 13.477 | 48.574 | 290 | 1826-1894 |

tury, our survey identifies 20 gauge stations with observation spans from 9 to 113 years; Regensburg is the longest continuous
series. 14 of these stations exceed 50 years of coverage, offering strong potential for long-term hydrological analysis (Table 1).

## 2.3 Case study

The charts function both as hydrological records and as administrative artefacts. Although layouts and production practices
are not fully uniform across years and stations, their broadly standardised format, dense daily annotation, and long temporal
coverage make them well suited to a semi-automated reconstruction workflow.

We analyse three representative gauges along the Bavarian Danube: Neu-Ulm, Vilshofen, and Passau, spanning the upper to
lower reaches of the stream within the country. Together they offer long temporal coverage and markedly different visual char-
acteristics (paper stock, handwriting, template variants), providing a robust and diverse test bed for semi-automated extraction.





## 3 Water-Level Reconstruction

### 3.1 Data Rescue from Historical Hydrographs

Despite their scientific value, much of this pre-digital record remains underused. Yet, climate and environmental data-rescue efforts emphasise that such sources are essential for extending baselines, quantifying long-term change, and improving extremes assessment (Brönnimann et al., 2018; Hawkins et al., 2023a). Long historical series enable retrospective flood chronology, attribution of anthropogenic impacts (land-use change, river training, canalisation), and independent validation of hydrological and hydraulic models used for planning and adaptation. They also support public communication of changing hazards by placing

recent events in a longer context (https://climatelabbook.substack.com/p/visualising-climate-impacts-on-uk). Recent reports underpin the importance of acquiring historical data (World Meteorological Organization, 2025). All these considerations are in line with single and coordinated efforts of data rescuing on all scales and in all areas of Earth System and Anthropocene Studies (e.g., *Copernicus*, "the European Union's Earth Observation Programme": https://datarescue.climate.copernicus.eu/, Huffman et al. (2023)). Especially in the context of water as an essential good, we hope to provide to a historical perspective

(WBGU, 2025a) to create a climate resilient strategic water management (WBGU, 2025b).

Recovering structured data from historical scientific graphics is variously termed *datafication of historical sources* (Donig and Rehbein, 2022; Rehbein, 2024; Navarro et al., 2025), *reverse engineering*, *data archaeology*, or *data rescue* (Poco and Heer, 2017; WMO, 2024). The aim is to reconstruct machine-readable series (with associated metadata) from pre-digital figures when the original tabular sources are lost, incomplete, or inaccessible. In hydrology, this typically means converting

hand-drawn hydrographs into daily or sub-daily level records with explicit uncertainty and datum control. For sea-level based data rescue, Latapy et al. (2023) give a comprehensive survey. Also seismology provides a closely related precedent: analogue seismograms on smoked paper or film have been digitised by tracing instrument curves into time series, with both manual and automated workflows now in use (Wang et al., 2019; Bogiatzis and Ishii, 2016; Paulescu et al., 2016; Aghajani et al., 2023; Wang et al., 2025).

In other fields such as medicine and chemistry, chart digitisation has advanced through shared benchmarks and competitions. ICDAR, for example, has stimulated progress in chart recognition and curve vectorisation by releasing annotated datasets and standardised evaluation tasks (Davila et al., 2019; Yepes et al., 2021). Extending similar frameworks to hydrology and geoscience would enable reproducible pipelines, shared ground-truth collections, and benchmarked performance. Yet these initiatives mainly address clean, modern prints and leave aside the distinctive problems of historical sources. Unlike

modern, digitally rendered plots with uniform backgrounds and crisp axes, the archival charts present heterogeneous artefacts that complicate line detection and coordinate recovery. The most relevant categories and their effects are (see Figure 5 for examples):

- **Page/scan geometry:** spine-induced curvature, local warp, minor tears, and quarter-join misalignment. These introduce slow baseline drift and small discontinuities that can bias the pixel→level conversion and day assignment;





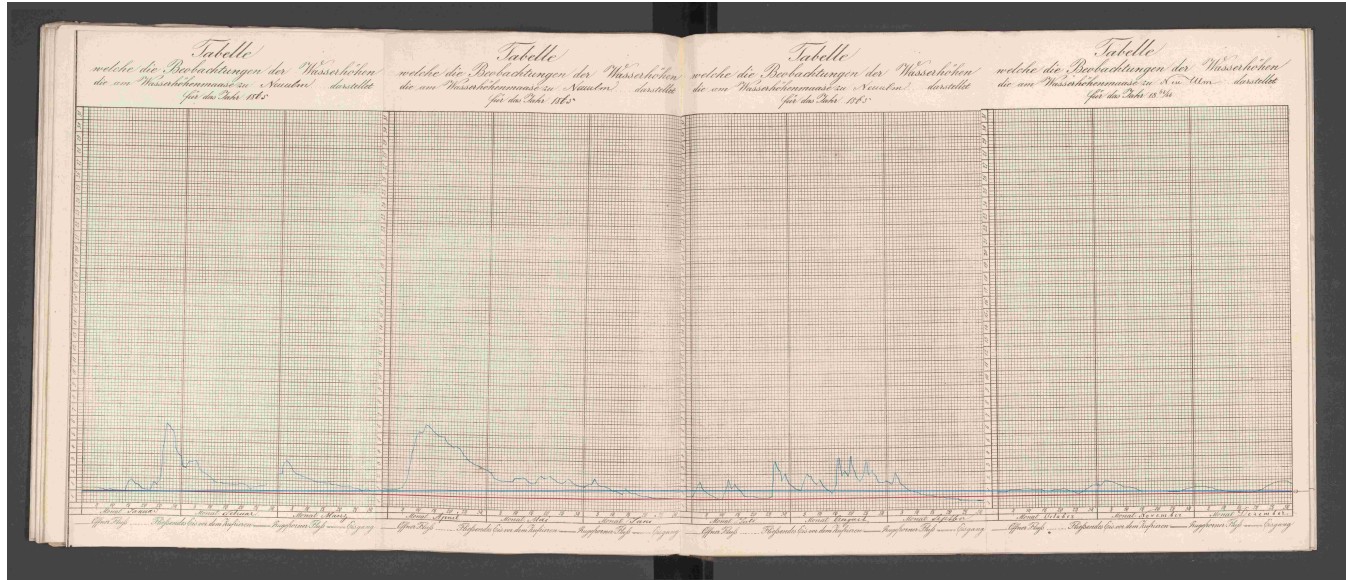

**Figure 4.** Book-spine-induced page curvature persists despite an orthogonally aligned scan. The blue line marks a constant $y$-pixel across the image; the red line indicates the form's zero-foot baseline. Around mid-year, near the spine bulge, the offset reaches about $\frac{3}{4}$ ft (21.9 cm) in gauge level and must be corrected during calibration (*Passau, 1865*). Source: BayHStA BLW 30.

– **Background and ink:** paper ageing, show-through, and ink bleed produce textured backgrounds; pigment loss (e.g. crayon rubbed thin) reduces stroke contrast. Both lower the signal-to-noise ratio and increase false positives;

    – **Grid and layout variability:** changes in grid alignment, vertical scaling, axis labelling, and template versions cause domain shifts that a model must accommodate; slight horizontal spacing inconsistencies can nudge date bins;

    – **Scribe/style effects:** variations in pen colour, line thickness, and drawing style across years and hands yield multi-modal
stroke appearances across stations, within a single station's series, and even within a station year;

    – **Editorial marks and anomalies:** sloppy drawing, later corrections, marginal annotations, occasional discontinuities of the curve, erroneous dates (e.g. 30 February), extra paper added for floods, and imperfectly glued quarters create distractors and small step offsets.

     Despite these challenges, the charts are broadly regular in layout and densely annotated at daily resolution, which makes
them well suited to *structured* extraction with transformer-based models when paired with light, grid-aware calibration and simple plausibility checks (details in Section 4). Figure 4 illustrates a common failure mode: book-spine-induced curvature that must be accounted for during calibration.



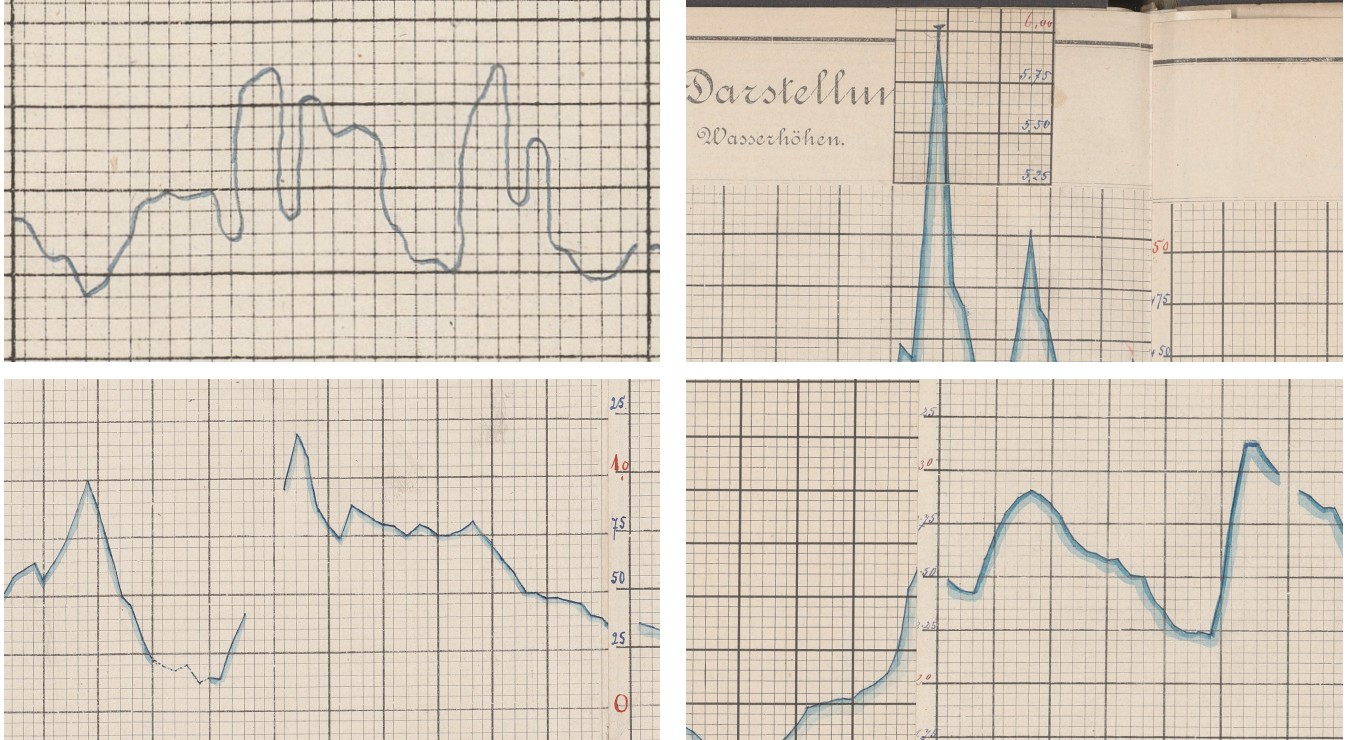

**Figure 5.** Samples illustrating some types of anomalies in line chart drawing often found in the sources. The selection also illustrates some of the different drawing styles. Top left: some creativity employed in this sample with the curve even going backwards (*Neu-Ulm, 1826*); top right: extra paper added for covering high flood (*Passau, 1892*); bottom left: non-continuous line between months February and March (*Vilshofen, 1894*); bottom-right: misaligned gluing of quarterly forms (*Passau, 1892*). Image sources: BayHStA BLW 21, 29, 30.

### 3.1.1 Problem structure

Across disciplines, the essential steps, whether conducted consecutively or in one go, are similar: (1) figure parsing (panel segmentation, axes and grid localisation), (2) coordinate system recovery (mapping pixels to physical units under a declared datum/zero), (3) trace extraction (identifying and vectorising the plotted curve), and (4) transform vectors into data using the coordinate system. Each step interacts with documentary conventions: changing templates, variable grid spacing, marginal notes, and later corrections. For hydrometric charts, the additional requirement is *datum consistency* across months/years (gauge zero, units), so that reconstructed series can be compared across time. Community practices in sea-level work (e.g., revised local reference schemes) illustrate the importance of such cross-record unification.

### 3.1.2 River and tide hydrographs

River charts and tide charts present different signal and layout regimes. River hydrographs emphasise seasonal cycles and episodic floods, often with long panels and coarser vertical ticks; tide charts exhibit strong semidiurnal/diurnal periodicity and





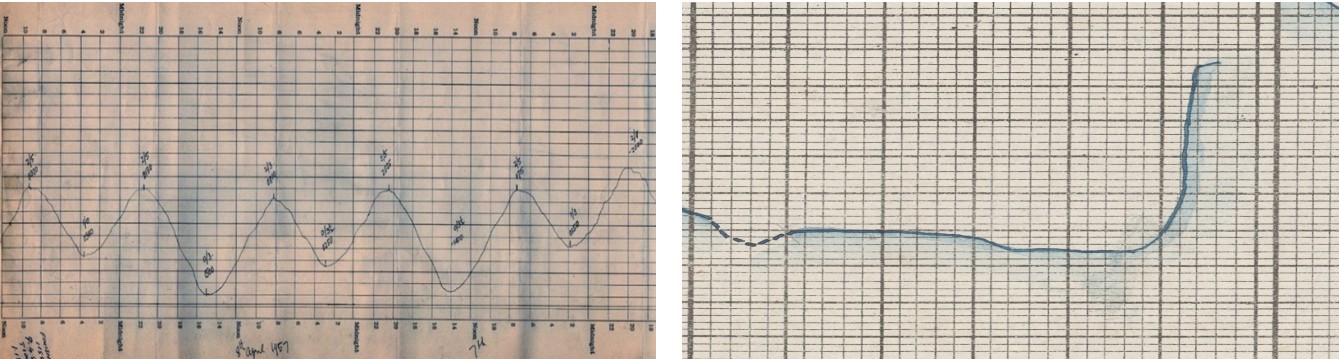

**Figure 6.** Juxta-position of a typical tidal sea level record (marigram) and a particularly flat period of a river water level. Left: Williamstown (Australia) tide data (1875–1965, here: 1957) illustrating typical periodicity. Source: Mcrae (2018); right: sample of a panel in river gauge level (*Neu-Ulm, February 1873*). Image source: BayHStA BLW 21.

denser vertical scaling (Latapy et al., 2023; Talke et al., 2020). In both cases, page artefacts (paper warp, spine curvature, quarter-join offsets), heterogeneous inks, and template revisions complicate automated extraction. The common core is a single, continuous hand-drawn line that must be followed through noise, gaps, and overprinted, written, or drawn annotations, but especially the panels/plateaus as visualised here caused additional challenges in differentiating them from the underlying grid (Figure 6).

## 3.2 Regional surveys of historical water-level data rescue

### 3.2.1 Americas

Across the Americas, tide- and river-level reconstructions combine archival discovery, datum control, and digitisation. NOAA's Joint Archive for Sea Level (JASL) provides a backbone for nineteenth-century tide data, with scanned marigrams and homogenised hourly records (Caldwell et al., 2001). Reviews identified large reserves of marigrams and registers and established workflows for recovery (Talke and Jay, 2013); NOAA's CDMP scanned thousands of tsunami charts. River data show similar promise: for the Lower Columbia, nineteenth-century stage records were digitised and merged with modern observations to assess long-term estuarine change (Helaire et al., 2019). These cases highlight archive scanning plus benchmark control as key to producing robust, machine-readable series.

### 3.2.2 Africa

African archives contain large volumes of undigitised meteorological logs, many dating back to the late nineteenth and early twentieth century. Recent work in the Democratic Republic of the Congo has scanned thousands of such records from 37 stations and applied machine-learning methods (*MeteoSaver*) to transcribe handwritten tables with high accuracy (Castelvecchi,



2025). These efforts illustrate both the scale of Africa's pre-digital heritage and the emerging role of AI in converting fragile paper series into usable long-term climate data.

### 3.2.3 Asia

Asia holds extensive analogue tide and river records with strong rescue potential. Around the Indian Ocean, recovery of marigrams has enabled re-analysis of early twentieth-century tsunami and tide events (Matsumoto et al., 2009). In India, long tide-gauge series such as Mumbai/Colaba, beginning in the late nineteenth century, have been re-examined with modern methods to assess long-term sea-level change (Unnikrishnan and Shankar, 2007). In South and Southeast Asia, river archives from the twentieth century (e.g. Mekong and major Indian basins) contain daily stage charts, and recent syntheses show how digi-

tised historical levels can underpin assessments of infrastructure impacts and regime shifts once station metadata and vertical references are unified. In South and Southeast Asia, river flow (notably the Brahmaputra basin) has been reconstructed over centuries to characterise long-term water-level changes, although through historical records but by utilizing dendrochonology (Rao et al., 2020). Similar examples working with archives of nature include the Ganges (Wasson et al., 2013) and the Mekong (Burnhill and Adamson, 2008).

### 3.2.4 Oceania

Australia and New Zealand have progressively rescued tide and river records by scanning, digitising, and tying them to geodetic networks. At Williamstown (Port Phillip Bay, Victoria), digitised registers and marigrams extended the local sea-level record by almost a century and were accompanied by open workflows (McInnes et al., 2024). Fremantle (Western Australia) illustrates the need for careful land-motion corrections, as geodetic studies revealed non-linear subsidence at the tide-gauge site (Featherstone

et al., 2015). At Port Arthur (Tasmania), mid-nineteenth-century observations (1841–1842), tied to a historic benchmark on the "Isle of the Dead," were recovered and merged with modern records to produce a continuous reference series (Hunter et al., 2003). In New Zealand, long-running digitisation efforts have combined historical gauges (Auckland, Wellington, Lyttelton, Dunedin) with geodetic control to quantify regional sea-level change (Hannah and Bell, 2012).

     Beyond tide gauges, Australia also hosts notable long-term hydrological series. Reconstructions of Murray River streamflow

extend variability estimates back to the late eighteenth century (Gallant and Gergis, 2011), while a two-century compilation of Lake George (NSW) water levels integrates diverse archival observations into a consistent reference series (Short et al., 2021). In New Zealand, archival staff-gauge readings, some extending to the late nineteenth century, document daily river levels prior to the advent of automated instrumentation (https://archivescentral.org.nz/horizons-regional-council/recordset/ hydrology-stage-%28water-level%29-daily-staff-gauge-readings). Collectively, these tide- and river-level initiatives demon-

strate a mature template for historical water-level reconstruction: scan and catalogue legacy sources, digitise with quantified uncertainties, unify vertical references through benchmark histories, and publish harmonised series with documented provenance.





### 3.2.5 Western Europe

European work combines data archaeology, datum control, and image-based digitisation. Frameworks emphasise marigram/register rescue and harmonisation (Woodworth et al., 2016; Latapy et al., 2023; Khan et al., 2023). Case studies show accurate chart-to-series conversion: Dún Laoghaire marigrams in Ireland (McLoughlin et al., 2024), Williamstown (McInnes et al., 2024), River Ouse (Macdonald and Black, 2010), and Thames ledgers (Inayatillah et al., 2022). Longer records (e.g., Liverpool, British Isles) highlight the need for careful quality control (Woodworth and Blackman, 2002; Bradshaw et al., 2016). For rivers, scanning of mechanical charts plus metadata curation extends sparse digital records; Irish catchment work offers replicable methods (De Smeth et al., 2024). Document restoration, warping correction, and datum management are essential to produce robust series.

### 3.2.6 Central Europe

Central Europe's long tide- and river-gauge traditions have yielded many successful rescues. Along the German Bight and Baltic, analogue records (e.g., Cuxhaven, Wismar) were scanned, datum-checked, and homogenised for trend analysis (Dangendorf et al., 2013, 2014; Kelln et al., 2019). Documentary and instrumental evidence for the Elbe and Oder provide multi-century chronologies (Becker and Grünewald, 2003). Curated datasets consolidate metadata for reproducible re-engineering (Kelln et al., 2019). Datum unification via PSMSL RLR and GNSS benchmarks remains central (Holgate et al., 2012). River archives likewise benefit from careful metadata curation and image-based line extraction (Becker and Grünewald, 2003). Complementary European frameworks document end-to-end procedures for digitisation and harmonisation (Latapy et al., 2023; Inayatillah et al., 2022).

### 3.2.7 Danube

The Danube combines documentary sources (flood marks, accounts, newspapers) with institutionalised hydrometry. Austria's *Hydrographisches Jahrbuch* (from 1893) provides daily to sub-daily summaries now digitised and online, anchoring older records to station datums. In Vienna, nineteenth-century regulation and dense documentation contextualise gauge data, as engineering works introduced datum shifts (Winiwarter et al., 2013; Hohensinner et al., 2013a, b). For the Middle Danube, compilations of medieval–modern floods situate nineteenth-century gauge data in a long-term context (Kiss and Laszlovszky, 2013).

### 3.3 Data Extraction: Methods and Practice

Several strategies have emerged to extract data from analog scientific figures, each with trade-offs in scalability, accuracy, and labour intensity. In the following, we briefly discuss manual annotation, citizen science, computer vision and machine learning, and transformer-based approaches.



### 3.3.1  Manual Annotation

The most direct approach of graphical charts, is manual annotation, using specialised software tools such as TIITBA Corona-Fernandez and Santoyo (2023), which provides interfaces for tracing, vectorising, and correcting seismogram curves. While
reliable, this process is time-consuming and difficult to scale. For large archives such as river gauge records spanning decades, fully manual workflows are prohibitive.

Manual (and semi-manual) digitisation remains a baseline strategy for re-engineering analogue charts and ledgers when automation is unreliable or when provenance requires human verification. Typical workflows comprise high-resolution scanning; image pre-processing (deskewing, dewarping); segmentation; axis calibration to the chart's printed grid; and interactive trac-
ing or point-picking along the plotted curve, followed by export to a regular temporal grid and documentation of datum/unit conversions (Mitchell et al., 2019).

In sea-level work, manual and semi-automated digitisation have recovered long, high-frequency series directly from paper marigrams—for example Dún Laoghaire (Ireland, 1925–1931), where careful geometric correction and uncertainty quantification were integral to the workflow; Belfast Harbour (UK, 1901–2010), which used scanning plus line-seeking software to
reconstruct a century-scale record; and Socoa/Saint-Jean-de-Luz (France), where a data-archaeology programme combined scanning, metadata curation and quality control to extend the series back to 1875 (McLoughlin et al., 2024; Murdy et al., 2015; Khan et al., 2023). For ledger-based archives (e.g. Thames high/low waters), manual transcription yields long homogeneous series when coupled with rigorous checks and clear provenance (Inayatillah et al., 2022). Reviews of sea-level data archaeology emphasise that analogue-to-digital recovery should document benchmark ties, instrument changes, and vertical datums
alongside the digitised series (Bradshaw et al., 2015; Latapy et al., 2023).

Manual digitisation offers transparency and expert control at page level and can achieve high accuracy on clean charts, but it is labour-intensive and sensitive to operator drift. Best practices therefore include (i) double-keying or replicate tracing for a sample of pages and (ii) inter-coder checks to quantify bias (Drevon et al., 2017). Where feasible, semi-automated toolkits tailored to marigrams (e.g. NUNIEAU) can accelerate tracing while preserving quality control (Ullmann et al., 2005).

### 3.3.2  Citizen Science Initiatives

Citizen science (CS) has become a practical accelerant for mobilizing historical sources in general (Rehbein and Ernst, 2023) and for re-engineering analogue environmental records at scale in particular (Navarro et al., 2025). Successful campaigns decompose digitisation into micro-tasks (e.g. transcription of register entries, tracing of plotted lines, identification of unit changes), replicate inputs across multiple volunteers, and fuse results by consensus with expert review. In this way, CS delivers
high-value datasets while preserving provenance and audit trails.

Large programmes have mobilised volunteers to transcribe historical meteorological records, including the UK&Ireland Rainfall Rescue (Hawkins et al., 2023b; Hawkins, 2025), *Unearthing Australia's climate history* (https://climatehistory.com.au/), Canada's *DRAW* project (https://draw.geog.mcgill.ca/), and *Jungle Weather* (https://www.zooniverse.org/projects/khufkens/jungle-weather). Long-running efforts initiated by *Old Weather* (https://www.oldweather.org/) continue to recover ship log





observations of pressure, temperature, wind and sea state (Brohan et al., 2009; Hawkins et al., 2019; Burr et al., 2021; Westcott et al., 2021; Teleti et al., 2024), while regional campaigns (e.g. Southern Weather Discovery, Antarctic logbooks) show how CS can target specific archives (Lorrey et al., 2022; Lakkis et al., 2023). For a broader overview, see https://datarescue.climate.copernicus.eu/citizen-science.

In Germany, the NFDI4Earth incubator pilots explicitly invite volunteers to datafy historical river gauge records, turn-
ing scanned hydrographic charts into structured level series (https://git.rwth-aachen.de/nfdi4earth/pilotsincubatorlab/incubator/do-it). Here, CS tasks extend beyond transcription: participants help identify axes and baselines, flag warped or damaged pages, and trace short line segments where automated extraction fails. Such experiments demonstrate how CS can be integrated directly into hydrographic data rescue, providing training and validation data for computer-vision models while ensuring quality control through human oversight.

Effective CS re-engineering projects typically (i) define unambiguous micro-tasks with visual examples; (ii) use n-fold replication with majority/weighted consensus; (iii) seed "gold-standard" items to measure volunteer accuracy and provide training feedback; (iv) surface low-agreement items for expert decision; and (v) maintain open, versioned data releases. For graphical sources (like gauge charts), CS tasks can include locating axes and baselines, flagging months with warping or template changes, and tracing short segments where automatic line-following fails.

CS scales human attention and can rapidly unlock otherwise inaccessible archives, but throughput depends on participant engagement and careful task design. Although automation is the ultimate goal in data rescue, CS may complement the process when manual attention is still required: for (a) generating training and validation data for machine-learning models (e.g. pixel-level line annotations, baseline marks), although synthetic data generation might be a viable option, too; (b) metadata enrichment (e.g. unit changes, marginal notes, year–page mapping); and (c) targeted quality control of months flagged as
ambiguous by the predictor. This hybrid strategy balances speed with reliability and reduces expert time spent on routine corrections.

### 3.3.3 Computer Vision and Machine Learning

Automated extraction from analogue charts has evolved from classical computer-vision pipelines to transformer-based systems that can detect chart components, reconstruct data tables, and reason about values. Early pipelines relied on edge detection,
Hough transformation, and active-contour (snake) models to trace plotted lines and grid structures. While effective on clean, modern figures, they are brittle on historical scans with ageing, noise, and warp (Kass et al., 1988). Interactive hybrids combine chart-type classification with semiautomated mark extraction and user corrections, improving robustness on heterogeneous material (Savva et al., 2011; Jung et al., 2017).

Recent deep-learning systems target "chart-to-table" derendering and chart understanding. *ChartOCR* introduced a hy-
brid framework that detects chart type and structural elements (axes, legends, lines) to reconstruct underlying values, and remains a strong baseline for line-chart data extraction (Luo et al., 2021). *ChartReader* unifies derendering and comprehension with transformer components, reducing heuristic rules and improving cross-task generalisation (Cheng et al., 2023). Question&Answer-style benchmarks such as *PlotQA* and *ChartQA* have driven progress on detection and reasoning over



plots and provide large synthetic/curated corpora for pretraining and evaluation (Methani et al., 2020; Masry et al., 2022).
Vision-language approaches like *DePlot* translate chart images into linearised tables that a language model can then query;
while domain adaptation is needed for historical sources, the modality-conversion step is directly relevant to time-series recovery (Liu et al., 2023).

For historical gauge charts in general (though not in our case study), accurate recovery hinges on robust polyline extraction
under noise, bleed-through, and curvature. Line-segment detectors and wireframe parsers (e.g. Holistically-Attracted Wire-
frame Parsing, HAWP) offer vectorised outputs with junction consistency and have been adapted to noisy, low-contrast imagery
(Zhao et al., 2020; Xue et al., 2023).

Paper bulging from book spines and scanning skew introduce systematic geometric error that propagates to level estimates.
Learning-based document dewarping—*DocUNet* and *DewarpNet*—predicts pixel-wise forward mappings from warped to flat
coordinates, offering a principled pre-processing step before line extraction (Das et al., 2019). In our context, mild, grid-aware
dewarping combined with per-month anchor calibration (Section 4.5.3) reduces day-shift and vertical bias.

Because annotated historical river charts are scarce, most systems rely on synthetic training corpora with realistic degra-
dations (ink bleed, paper texture, yellowing, local warp) and self-supervised pretraining. Public chart datasets (e.g. *PlotQA*,
*ChartQA*) and chart-derendering corpora (*ChartOCR/ChartReader*) provide pretraining targets; fine-tuning on even small,
well-curated hydrological samples markedly improves stability.

### 3.3.4   Transformer-Based Models

Transformers have become the workhorse for chart derendering and curve extraction because self-attention captures global
context (axes, legends, gridlines) while retaining fine-grained detail along thin strokes.

End-to-end detectors in the *DETR* (DEtection TRansformers) lineage treat chart components as a set prediction problem,
reducing post-processing and improving robustness to clutter (Carion et al., 2020). For line charts, LineFormer, which we build
on in the following, reframes data extraction as instance segmentation of polylines and reports strong performance on synthetic
and scanned material; its decoder naturally separates overlapping traces and simplifies downstream coordinate mapping (Carion
et al., 2020; Lal et al., 2023).

Vision–language transformers *Donut* (Document Understanding Transformer) and *Pix2Struct* parse documents and UI-like
artefacts directly to token sequences, while DePlot converts plots to linearised tables for subsequent reasoning by an LLM.
*UniChart* pretrains on diverse chart corpora to support QA and summarisation. These models have been developed for modern
graphics and not yet tested for adaption to historical hydrological samples (Kim et al., 2022; Lee et al., 2023; Liu et al., 2023;
Fields and Kennington, 2023; Masry et al., 2023).

Overall, transformer-based models, such as *LineFormer* which we chose for our study, offer a high-leverage starting point
for nineteenth-century gauge charts.





## 4  Method of Reconstruction

### 4.1  Goals and design

The primary objective of this study is to convert nineteenth-century Bavarian Danube water-level charts into daily gauge-level series expressed in physical units (millimetres), referenced to the respective gauge zero. Beyond the technical reconstruction itself, the approach is guided by three overarching goals: (1) efficiency, i.e. minimising the need for manual intervention; (2) effectiveness, i.e. ensuring completeness and accuracy, and (3) reliability of the reconstructed data, i.e. providing a transparent, evaluated, reproducible and well-documented workflow.

The research design follows a stepwise structure aligned with these goals. First, evaluation criteria are defined to assess accuracy and practicability. Second, a representative sample is selected and a ground-truth dataset established. Third, an algorithm for pixel-to-curve transformation is formed, followed by an experimental pilot study to test the workflow. This includes image pre-processing, warping adjustments, and the optimisation of LineFormer hyperparameters. The results are benchmarked against the ground-truth data. Fourth, systematic errors are analysed and the workflow iteratively adjusted. Finally, the optimised workflow and parameters are subjected to final benchmarking and validation, forming the basis for subsequent large-scale production.

### 4.2  Evaluation protocol

This section specifies how we quantify the accuracy of reconstructed daily gauge levels. Our choices are shaped by two constraints of the corpus: (i) evaluation is conducted *per month* (the operational unit for extraction and quality control), and (ii) we do not rely on pixel-level ground truth polylines for accuracy; instead, we evaluate against ground truth in original physical units. Consequently, curve-space distances (e.g., Hausdorff/Fréchet) are not used as headline measures. We do, however, report a panel of series-level metrics—RMSE, MAE (average deviation), maximum absolute deviation, `Pearson_r`, and an adopted *peak-aware composite score* as our primary objective for decisions in the pilot study (parameter selection) and subsequent evaluations.

#### 4.2.1  Unit of analysis and dataset separation

Evaluation is conducted on a month-by-month basis; we never split within a month or use partial page fragments. Sampling is stratified across gauges and years to balance coverage; no model training is performed during these evaluations.

Hydrological use cases may care about physical levels and timing at daily resolution (flood peaks, seasonal lows). Series-level metrics in original physical units directly reflect those targets and are robust to purely graphical idiosyncrasies (ink density, micro-warp) that do not affect the numerical series. Pixel-space curve distances would require an additional layer of ground truth and can over-emphasise small geometric deviations that have negligible effect after conversion to levels. For this reason we (i) evaluate in level space, and (ii) reserve pixel-space diagnostics (continuity/fragmentation, edge hits) for internal quality control (by visual inspection, especially in candidate selection) rather than accuracy reporting.



### 4.2.2 Base metrics (series-level)

Let $(y_1, \ldots, y_N)$ be the ground-truth daily levels (in millimeters) for a given month $m$, $(\hat{y}_t, \ldots, \hat{y}_N)$ the predicted levels after conversion from pixels in the scanned images, and $N \in \{28, 29, 30, 31\}$ the number of days of $m$. We then define the difference, i.e. the error between a data point in the prediction series and the ground truth, for a given date (day) $t$ as $e_t = \hat{y}_t - y_t$.

**Root mean squared error (RMSE).** We consider

$$\text{RMSE} = \sqrt{\frac{1}{N} \sum_{t=1}^{N} e_t^2} \tag{1}$$

a standard overall accuracy in data series comparison; squaring emphasises larger mistakes (e.g., missed peaks) and expresses error in physical units. While it is sensitive to outliers, it can be dominated by a few large errors; less interpretable when errors are highly skewed.

**Mean absolute error (MAE) / average deviation (AD).**

$$\text{MAE} = \frac{1}{N} \sum_{t=1}^{N} |e_t| \tag{2}$$

reports the "typical miss" in actual units (millimeters), it is the average deviation (AD) between two series. While it is robust and easily comparable across the corpus as it is in the same unit as the gauge levels itself, it is less sensitive to rare but important extreme errors.

**Maximum absolute deviation (MaxAE).**

$$\text{MaxAE} = \max_{1 \leq t \leq N} |e_t| \tag{3}$$

reports the worst-case point error over the month. It is useful because it explicitly guards against single catastrophic misses, but it ignores overall performance and can be noisy month-to-month. Both, MAE and MaxAE indicate the potential impact of errors as they are given in gauge level unit.

**Pearson correlation coefficient (`Pearson r`).** Pearson $r$ measures co-variation of shape: $r$ is invariant to affine changes of either series (additive offsets and positive rescalings), so it reports agreement in the pattern of rises and falls independently of datum or overall amplitude. It is also dimensionless and bounded in $[-1, 1]$, hence directly comparable across stations, years, and unit systems. Pearson $r$ is widely used in time-series validation as a complementary view to magnitude-based errors (RMSE/MAE), helping to separate shape fidelity from bias and scale issues.

$$r = \frac{\sum_{t=1}^{N} (y_t - \bar{y})(\hat{y}_t - \bar{\hat{y}})}{\sqrt{\sum_{t=1}^{N} (y_t - \bar{y})^2} \sqrt{\sum_{t=1}^{N} (\hat{y}_t - \bar{\hat{y}})^2}}, \qquad \bar{y} = \frac{1}{N} \sum_t y_t, \; \bar{\hat{y}} = \frac{1}{N} \sum_t \hat{y}_t. \tag{4}$$

A high $r$ confirms that peaks, recessions, and seasonal phases co-vary correctly even when an overall offset or gain error exists. This makes $r$ particularly useful during pre-processing and datum control: combinations that preserve $r$ while harming RMSE/MAE typically introduce bias/scale errors rather than distorting the hydrograph.




### 4.2.3 Peak-aware composite metric

We judge the hydrograph reconstruction by how well they reproduce both the overall shape and the seasonal/flood peaks in value and timing of the original. To capture these aspects in a single indicator, we adopt a custom peak-aware metric that combines and weighs three components:

1. **Pearson correlation:** $r$, as above. If either series is constant/invalid, we set $r := 0$ for stability.

2. **Peak value similarity:** Let $y_{\max} = \max_i y_i$ and $\hat{y}_{\max} = \max_i \hat{y}_i$. We define *peak value similarity* as:

$$
s_{\text{peak-val}} = \begin{cases} 1 - \dfrac{|y_{\max} - \hat{y}_{\max}|}{y_{\max}}, & y_{\max} \neq 0, \\ 0, & y_{\max} = 0 , \end{cases}
\tag{5}
$$

which equals 1 for an exact peak match and decreases linearly with relative peak error. *Note:* $s_{\text{peak-val}}$ is not clamped above or below $[0,1]$; large over-prediction can make it negative. We implemented this behaviour to preserve sensitivity to extreme overshoots.

3. **Peak timing alignment:** Let $i^\star = \arg\max_i y_i$, $j^\star = \arg\max_i \hat{y}_i$, $d = |t_{i^\star} - t_{j^\star}|$ (days). We define *peak timing alignment* as:

$$
s_{\text{peak-time}} = \max\left( 0, \, 1 - \frac{d}{N} \right).
\tag{6}
$$

This lies in $[0,1]$ by construction (1 for same-day peaks, 0 when the shift spans the month length). It penalizes delayed prediction: the closer to 0, the later the predicted peak was from the actual one.

The peak-aware composite metric is computed on aligned daily series after an inner join on dates. With user-chosen $\alpha, \beta \in$ 405 $[0,1]$ and $\gamma = 1 - \alpha - \beta$, we define our custom peak-aware composite metric as:

$$
\text{score} = \alpha \, r + \beta \, s_{\text{peak-val}} + \gamma \, s_{\text{peak-time}}
\tag{7}
$$

The upper bound of score is 1; the lower bound can be $< 0$ if $r = -1$ and/or $s_{\text{peak-val}} < 0$. When $\alpha + \beta \leq 1$, the terms form a convex combination; when $\alpha + \beta > 1$ (thus $\gamma < 0$), the timing term reduces the score—useful if timing errors must be penalised more heavily than value errors. The default choice in our reference implementation is $\alpha = 0.4$, $\beta = 0.4$ (thus $\gamma = 0.2$), which 410 is a peak-sensitive setting emphasising overall shape and value of the curve without disregarding timing.

RMSE and MAE capture magnitude accuracy; `Pearson r` captures shape; none alone reflects event significance (peaks). The composite explicitly encodes the three desiderata—shape, peak amplitude, and peak timing—in a single, tunable quantity. This reduces multi-criterion ambiguity when selecting parameters in the pilot study (e.g., stretch factor, baseline aid, resampling rule): we can optimise a single objective that aligns with hydrological priorities, then audit trade-offs with the base metrics.





### 4.3 Source Preparation and Metadata Acquisition

**Volumes and page–year mapping.** The original gauge level documentation is preserved by the Central Bavarian State Archive (*Bayerisches Hauptstaatsarchiv, BayHStA*) in bound volumes, typically one per gauge, covering long multi-decadal periods (1826–1894 in our corpus). Each year is assembled from four quarterly forms that were glued edge-to-edge into a single long slip before binding. Non-chart pages (notes, memoranda, corrections) are interleaved irregularly. Consequently, scan/page indices are not monotonic in calendar time. As a first step we compiled a book-level table of contents that maps each document identifier and page number to a unique (gauge, year) pair and flags pages without charts. This register underpins sampling and all subsequent tasks.

**File organisation and provenance.** Scans are used by the archival identifier schema: ProvenanceID_GaugeIDPageID (e.g. Bay_Landesamt_fuer_Wasserwirtschaft_210085 for the records from Neu-Ulm, 1891). For each image we create a metadata record with: (i) gauge identifier, name and image size; (ii) nominal year; (iii) unit system noted on the form (Bavarian foot/mm); and (iv) page-level notes (annotations, overwriting, template changes). This ensures page-level provenance and allows exact reproduction of any result.

**Metadata Extraction.** Each scan was linked to its calendar year and measurement unit by decoding the archival identifier and inspecting marginal notes. Interleaved occasional commentary pages required manual checks to ensure that the image–year mapping is unambiguous. Axis units change from Bavarian feet to millimetres on 1 April 1872 (Figure 7); this transition is flagged in the pipeline and propagated to the coordinate-to-level conversion routines.

To complement the coordinate data, we manually transcribed the commentary pages for the entire document series, extending beyond the sample itself. This enriches the human contextual understanding of each chart and provides additional semantic layers for future modelling.

### 4.4 Ground-truth dataset construction

Reliable ground-truth data are essential for assessing the accuracy of predicted water-level curves and for calibrating the pixel-to-gauge transformation. We compiled two complementary ground-truth sets: (i) pixel-level annotations of the chart line and (ii) manually keyed gauge-level values. Together they form the basis for evaluation and pipeline optimisation.

#### 4.4.1 Sample selection

Fifteen annual charts were randomly sampled from three Danube gauges—Neu-Ulm, Vilshofen and Passau—yielding five charts per site (Table 2). This stratified selection balances spatial coverage with manageable annotation effort and captures variation in handwriting, paper quality and measurement units.



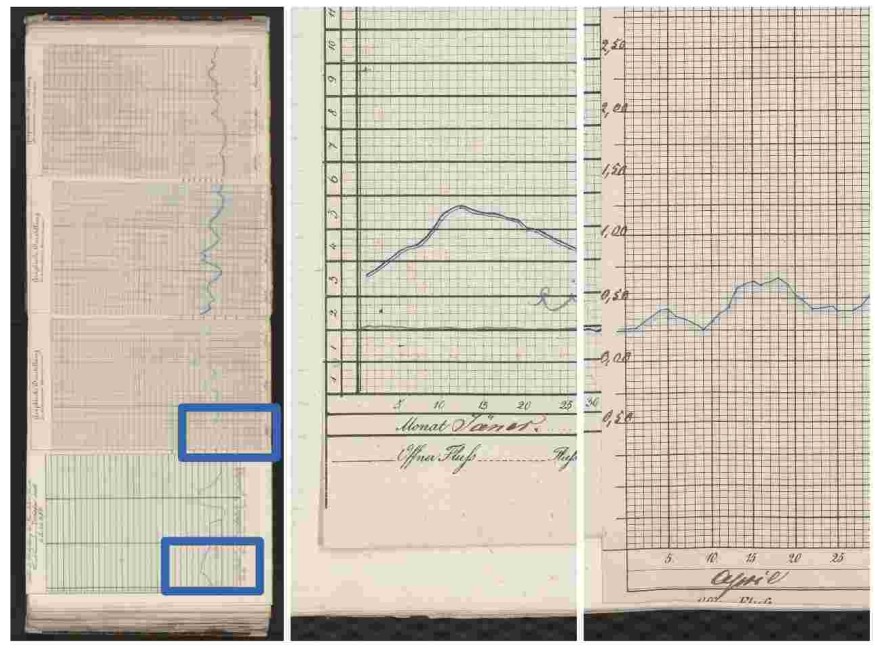

**Figure 7.** The Vilshofen 1872 chart illustrates the unit change from Bavarian foot to millimeters effective 1st of April. Note that at the transition from March to April, the chart aligns. Images source: BayHStA BLW 21.

**Table 2.** Sample random selection.

| Gauge location | DocID[a] | Pages | Years |
|---|---|---|---|
| Pegel Neu-Ulm | 21 | 18, 45, 51, 56, 85 | 1839, 1865, 1871, 1876, 1891 |
| Pegel Vilshofen | 29 | 22, 24, 45, 53, 69 | 1844, 1846, 1866, 1872, 1883 |
| Pegel Passau | 30 | 26, 46, 49, 60, 70 | 1848, 1865, 1868, 1878, 1888 |

DocID and page number form the full document identifier as follows: Bay_Landesamt_fuer_Wasserwirtschaft_{DocID}{Page:04}, e.g., Bay_Landesamt_fuer_Wasserwirtschaft_210018.

### 4.4.2 Pixel-based ground truth

**Annotation workflow.** Using an in-house tool `point_annotator` we traced the gauge curve at daily resolution (1 January
– 31 December). Each vertex stores pixel coordinates (x, y) relative to the full-page scan; an experienced annotator required ≈
12–15 min per annual chart (see Figure 9 for a sample outcome).

**Temporal mapping.** The ordered list of vertices is mapped to calendar days under the assumption of uniform horizontal
spacing within each month; leap years are detected automatically.





**Figure 8.** Annotation of y-axis anchor points during monthly annotation using the `month_annotator` tool. Image source: BayHStA BLW 21.





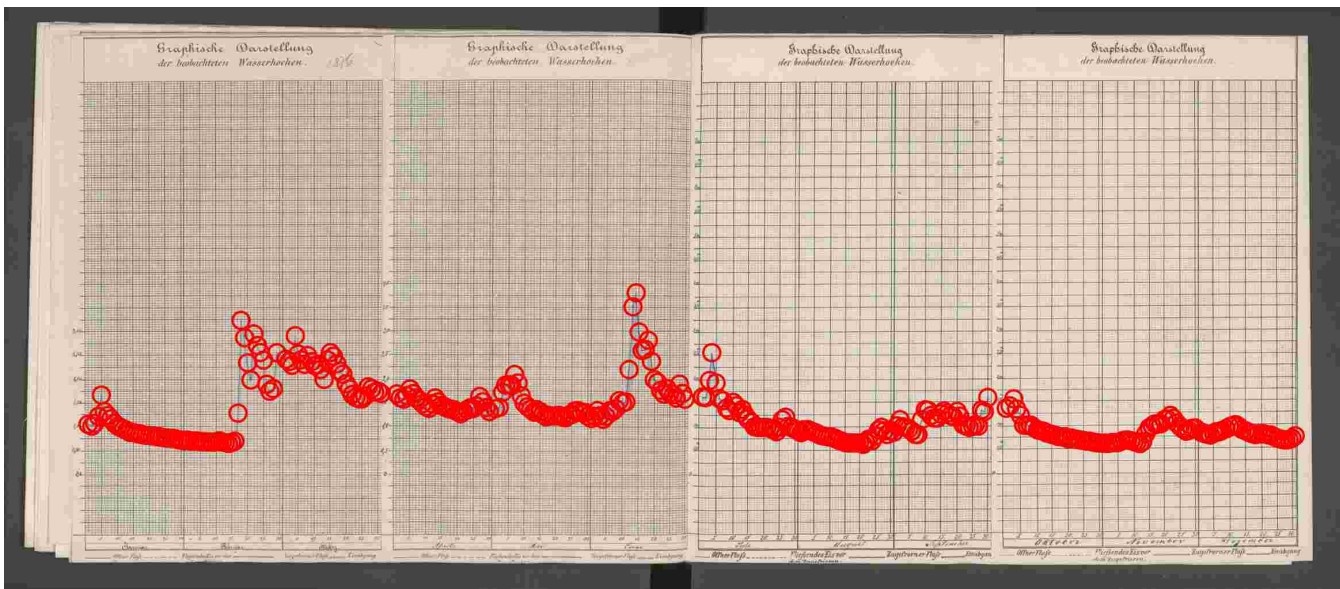

**Figure 9.** Sample of a pixel-based ground-truth annotation (Neu-Ulm, 1876). Image source: BayHStA BLW 21.

**Vertical scaling.** Already during the annotation for the monthly slicing, two anchor points ($c_{\text{low}}$ and $c_{\text{high}}$) on the y-axis were
recorded per chart in pixels and the according gauge level values manually entered in the chart's unit (foot or mm: $v_{\text{low}}$ and
$v_{\text{high}}$). Assuming linear scaling, physical gauge levels $v_{\text{test}}$ are recovered from pixel y-coordinates $c_{\text{test}}$ via

$$v_{\text{test}} = \frac{\left(c_{\text{test}} - c_{\text{low}}\right)\left(v_{\text{high}} - v_{\text{low}}\right)}{c_{\text{high}} - c_{\text{low}}} + v_{\text{low}} \tag{8}$$

### 4.4.3 Level-Based Ground Truth

For 4 of the same 15 charts we independently keyed the gauge level for each day directly from the grid ($\approx$70 minutes per
chart). The two ground-truth sets agree to 0.996 on Custom peak-aware score (see below) with a maximum absolute deviation
of 2.614 mm, demonstrating that pixel-based annotation is both reliable and an order of magnitude faster.

### 4.4.4 Cross-Validation and Quality Control

A random subset of transformed curves was visually overlaid on the original scans to detect residual misalignments caused by
scanning artefacts (warp, skew, page curvature). None exceeded one grid interval. These checks, together with the high agree-
ment between both annotation modes, give confidence that the ground-truth data are fit for purpose. All remaining distortions
are systematic and addressed later by the affine calibration described in Section 4.4.5.





As a third method for creating ground truth data (and also as an alternative benchmark for our semi-automated workflow), re-drawing the chart line on the screen using a stylus or graphic tablet with a post translation of coordinates as above is thinkable. This approach was not tested by us, but we may well assume a slight gain in efficiency without loss of quality.

### 4.4.5 Curve-to-Series Transformation

Since LineFormer predictions are denser than daily sampling (typically 2,000+ points per month), the selected line data is resampled to one point per day. We partition each month into 28–31 (depending on calendar length) equal slices and retain the last coordinate in each slice, which outperformed median, mean and medoid aggregations in our evaluations. X-coordinates are mapped to ordinal day numbers; y-coordinates are converted to physical gauge levels utilizing the per-tile anchor pair defined during the monthly slicing process using Eq. 8.

### 4.5 Experimental pilot study

The pilot phase determined workable inputs for transformer-based extraction on historical scans and locked key pre-processing choices. This phase was conducted with the sample (ground truth) data described in Section 4.4, i.e. 15 gauge years from 3 different gauges with an overall of 180 monthly data.

### 4.5.1 Pre-processing: slicing, cropping and y-axis calibration

Following the promising result on a single test image, we adopted a pre-processing strategy based on monthly horizontal slicing combined with vertical cropping. As previous experiments suggested that horizontal stretching might further enhance prediction quality, we proceeded to systematically evaluate its impact.

We evaluated full-page, semi-annual, quarterly and monthly tiling. Monthly tiles reduced local warp and date-assignment error and were adopted for the pipeline. As automating this step is reserved for future advances of the pipeline, full-page scans were manually annotated with the `month_annotator` tool to identify the chart area as bounding boxes for one month each. Precise placement proved critical: a small x-overshoot can shift the date assignment by a whole day (Fig. 10).

From the annotation, twelve horizontal tiles were automatically produced. The human annotation workload for this averaged 160 seconds per gauge year which includes the annotation of vertical anchor points required for y-axis transformation from pixels to gauge levels (see above, Section 4.4.2).

This monthly cropping had another effect. While its primary goal was to slice the yearly full image horizontally into man-ageable and processable units, it also allowed to reduce the vertical space of the image to process to what was actually needed by the graph. Eliminating an often high percentage of the space, made the task easier for the predictor.

### 4.5.2 Preprocessing: horizontal stretching

Monthly tiles are visually compressed along the horizontal (time) axis: steep strokes and tight meanders can collapse into only a few pixels per day. This increases local tangling and small gaps in the predicted polyline. A purely horizontal anisotropic





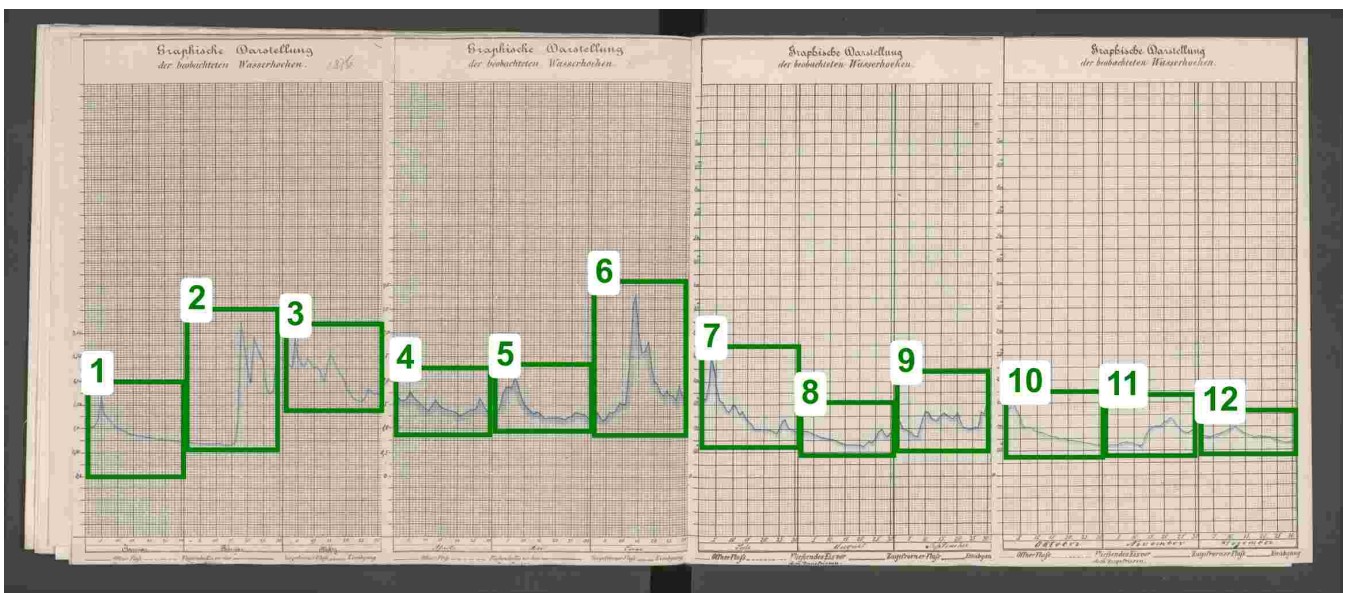

**Figure 10.** Annotation of monthly segments for slicing (Neu-Ulm, 1876). Image source: BayHStA BLW 21.

stretch increases the effective sampling along time without altering vertical geometry or units. In practice, stretching reduces steep slopes and widens close strokes, making stroke-following easier for the transformer while leaving the y-axis calibration (anchors, units) unchanged.

We evaluated different stretch factors $s \in \{0.5, 0.75, 1.0, 1.25, 1.5, 2.0, 2.5, 3.0, 4.0, 5.0\}$ applied to the x-axis only using bicubic interpolation (y-coordinates and vertical anchors unchanged). For each $s$, we re-ran inference on the development set (complete months) under identical post-processing: candidate limit set to 1, same day-slicing (last-in-bin), and the same pixel⟶level conversion. We then computed evaluation scores, especially *peak-aware Custom* against ground truth (Figure 11).

The quality improved monotonically from $s = 0.5$ to $s = 2.0$, with the largest gains between $s = 1.0 \rightarrow 1.5$ and diminishing returns thereafter. Beyond $s = 2.0$ the curves plateaued: average gains became negligible and several months showed small regressions. We therefore fix $s = 2.0$ for the production pipeline. For rare pages with atypical layout or memory constraints, $s$ may be changed, but all data produced here use $s = 2.0$. Vertical scaling and date binning is preserved, i.e. no change to y-units is performed and no month bounds affected.

### 4.5.3   Baseline adjustment

To mitigate spine-induced page curvature (relevant especially for the "inner" months of June and July, see Figure 4 for an example), we apply a light, grid-aware baseline adjustment using the printed zero line we manually annotated as a polyline with the tool `baseline_annotator.py`. The polyline serves as a normalization for the y-value-conversion, i.e. we subtract





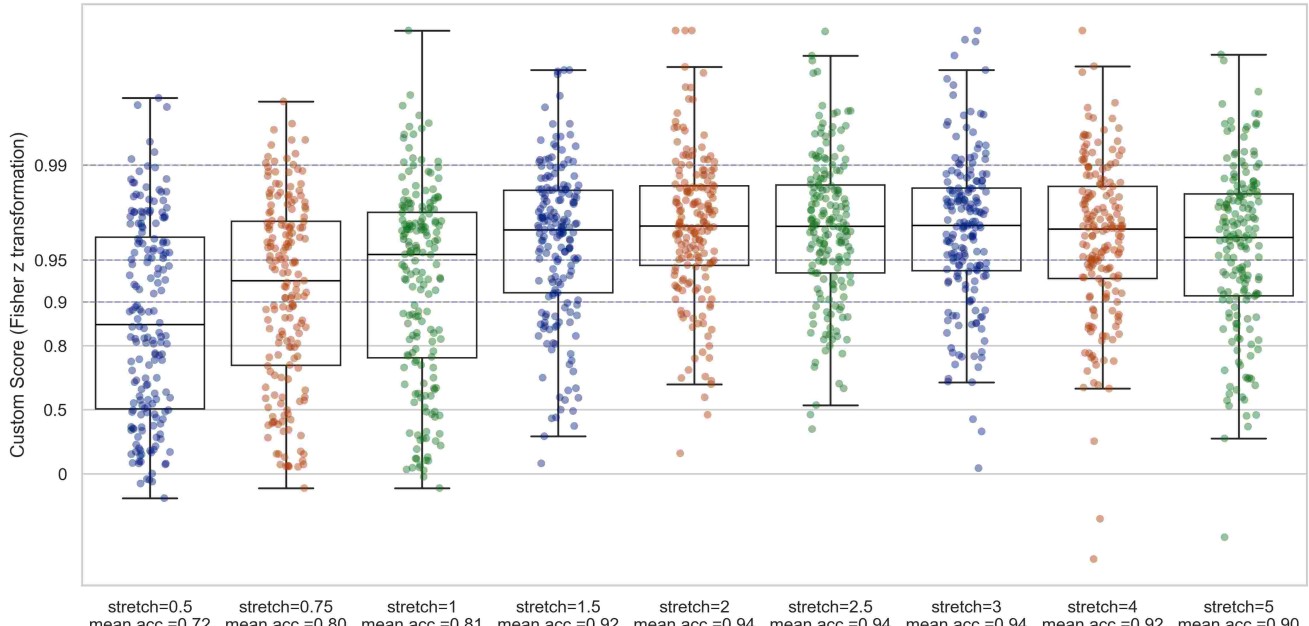

**Figure 11.** Evaluation of x-axis stretching factors against ground truth with peak-aware Custom metric.

locally estimated baseline bias before applying the linear y-scale Aligned with the horizontal image boundary, the distortion is
calculated and added to adjust the gauge level while converting from pixel values as follows.

Based on the calibration values from Section 4.4.2, let $\beta$ be the linear box correction as

$$\beta = c_{\text{low}} - \frac{v_{\text{low}}}{v_{\text{high}} - v_{\text{low}}} \left( c_{\text{high}} - c_{\text{low}} \right) \tag{9}$$

Let $((x_1, y_1), \ldots, (x_n, y_n))$ be an (x-)ordered set of pixel pairs to describe the annotated baseline polyline. For each segment
with endpoints $(x_i, y_i)$ and $(x_{i+1}, y_{i+1})$ and each $x_q$ with $x_i < x_q < x_i + 1$, let

$$\hat{y}(x_q) = y_i + (y_{i+1} - y_i) \frac{x_q - x_i}{x_{i+1} - x_i} \tag{10}$$

Then, the adjusted y-value $y(x_q)$ is calculated as:

$$y(x_q) = \hat{y}(x_q) - \beta \tag{11}$$

Tab.3 summarises the recognition accuracy comparing predictions with and without baseline adjustment. We observe a
general improvement in mean Custom peak-aware score which is (only) slightly higher in the critical central months. In terms
of efficiency, this additional manual pre-processing averages 26.89 seconds per page (gauge year) for annotating the polyline
base. Overall, we assess manual baseline adjustment as valuable and hence leave it in the pipeline.





**Table 3.** Accuracy (Custom peak-aware score) of sample predictions with and without baseline adjustment.

|  | count | mean | std | min | 25% | 50% | 75% | max |
|---|---|---|---|---|---|---|---|---|
| **without baseline adjustment: all** | 180 | 0.958 | 0.067 | 0.467 | 0.957 | 0.978 | 0.988 | 1.000 |
| **without baseline adjustment: June/July only** | 30 | 0.943 | 0.070 | 0.691 | 0.934 | 0.968 | 0.985 | 0.998 |
| **without baseline adjustment: all but June/July** | 150 | 0.961 | 0.067 | 0.467 | 0.961 | 0.980 | 0.989 | 1.000 |
| **with baseline adjustment: all** | 180 | 0.968 | 0.055 | 0.474 | 0.967 | 0.980 | 0.990 | 0.999 |
| **with baseline adjustment: June/July only** | 30 | 0.956 | 0.059 | 0.695 | 0.957 | 0.974 | 0.984 | 0.996 |
| **with baseline adjustment: all but June/July** | 150 | 0.970 | 0.054 | 0.474 | 0.968 | 0.981 | 0.990 | 0.999 |

### 4.5.4 LineFormer hyperparameters and candidate selection

Polyline prediction is based on the *LineFormer* implementation by Lal et al. (2023) which in turn builds on the open *MMDetection* framework (Chen et al., 2019). We made only minor adjustments to *LineFormer* to expose and analyse two scores,
*confidence* and *coverage*. Here, *coverage* denotes the fraction of the image's x-axis for which a data point is predicted. Because images were sliced to span the expected chart range (first to last day of the month), coverage should ideally approach 100%. We treat , coverage $> 0.985$, as viable, corresponding to a potential time bias smaller than half a day (i.e. less than one vertical grid interval), consistent with the manual visual inspection described below.

Among *LineFormer*'s hyperparameters, only `maxperimage`—the maximum number of polylines predicted per image
(gauge–month)—had a material impact on outcomes. The default is `maxperimage = 100`, which in our sample produces between 1 and 12 polylines per image (see Fig. 12 for an example). As our sources indicate that each image contains exactly one chart, `maxperimage = 1` would be ideal, as it obviates candidate selection. However, this setting does not always yield the most accurate or most complete prediction.

We evaluated `maxperimage` $\in 1, 2, 3, 100$ and compared accuracies. Whenever *LineFormer* returned more than one pre-
diction for an image (94.4% of cases), a selection step was required. For `maxperimage` $> 1$ we used the model's *confidence* score to select a single candidate. Values `maxperimage` $> 1$ did not improve accuracy relative to `maxperimage = 1`, making `maxperimage = 1` the only viable option for fully automated processing at this stage (see Fig. 13 and Tab. 4).

As an alternative to automatic selection by confidence, we conducted manual review with the `inspector.py` tool at `maxperimage = 100`. A human evaluator inspects all predictions per image, selects the visually best curve, and flags it as
requiring a major correction (deviation $>$ one grid tick on either axis), a minor correction (within one tick), or no material correction. This process takes approximately 3 s per gauge–month. In our 180-month sample, 88.9% of months were accepted without correction, 3.9% were flagged for minor edits, and 7.2% for major edits. Manual selection outperformed automatic confidence-based selection (overall accuracy 0.968 vs. 0.886).

Within the `maxperimage = 100` setting, we asked whether manual selection could be approximated by a simple com-
bination of confidence and coverage. Using confidence alone recalled 78.2% of the visually best predictions. GroupKFold



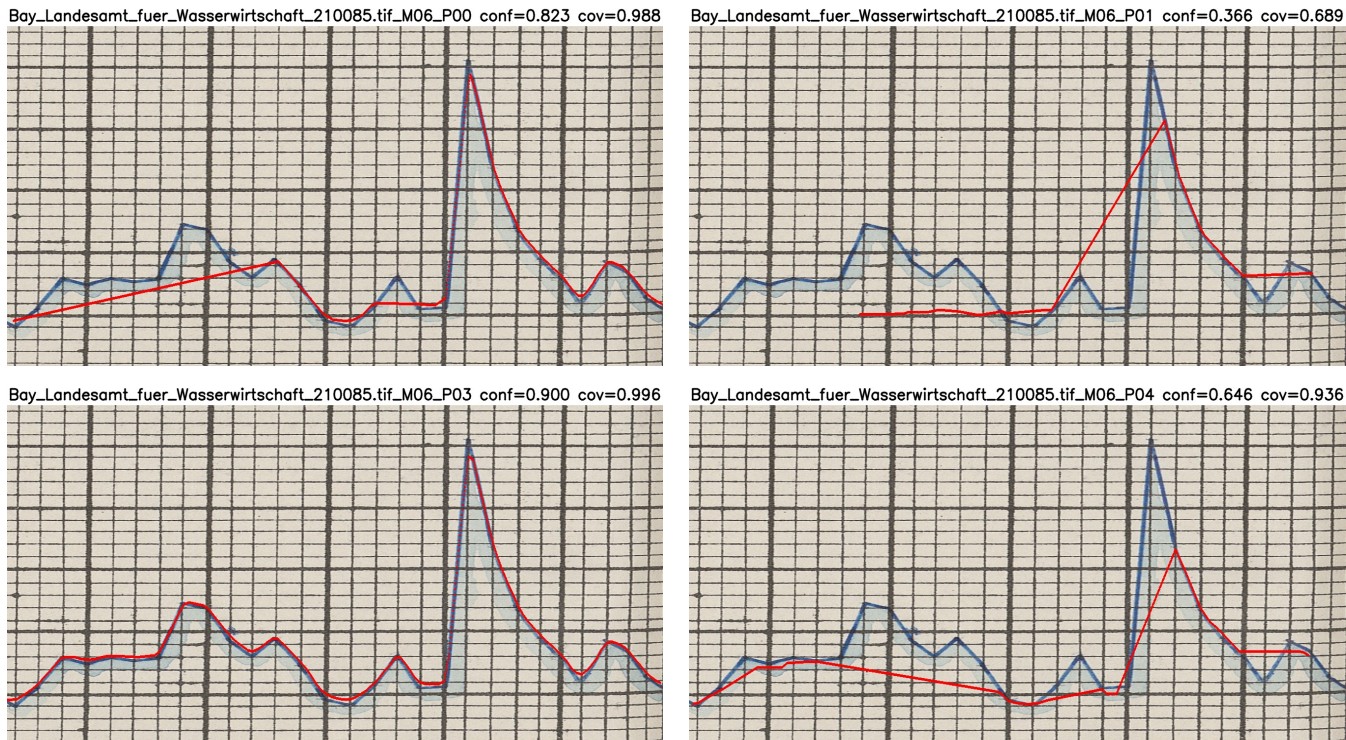

**Figure 12.** Samples illustrating different candidates returned by *LineFormer* during the same prediction run (Neu-Ulm, June 1891). Image source: BayHStA BLW 21.

cross-validation identified $\alpha = 0.69$ as the optimal weight in the convex combination

$$s_\alpha = (1 - \alpha) \cdot confidence + \alpha \cdot coverage \tag{12}$$

for automated candidate selection. This increased recall to 87.7% and improved overall accuracy from 0.886 to 0.937.

In summary, fully automated selection still underperforms manual choice by a meaningful margin (0.937 vs. 0.968). Manual inspection additionally helps to identify cases that benefit from post-prediction correction. We therefore retain manual visual inspection in the processing pipeline. As a pragmatic alternative, setting `maxperimage = 1` and adding a lightweight human flag for predictions requiring post-correction appears sound.

### 4.5.5 Post correction

Post correction was conducted and evaluated in those ca. 11% cases for which the human inspector decided that minor or major deviations exist. As seen above, it improved the mean accuracy significantly from 0.968 to 0.979. In terms of efficiency, however, this manual correction with tool `corrector.py`, which allows movements of single points on a graphical display, comes at a cost of averaging 60 sec. per gauge month.





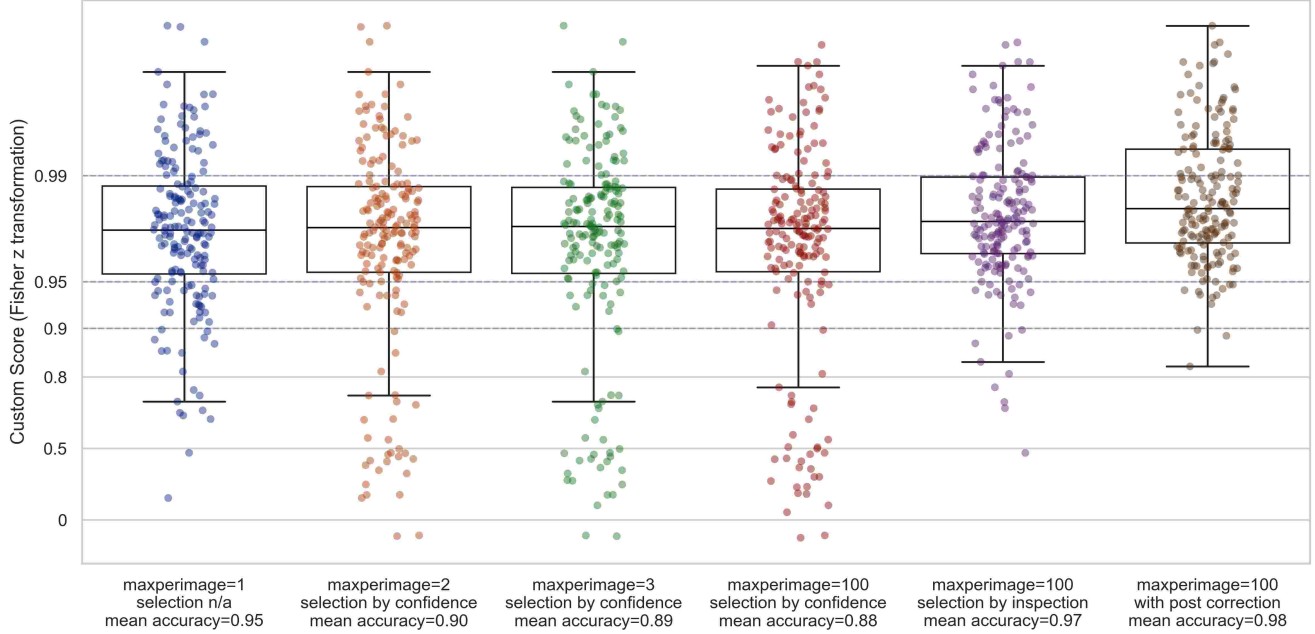

**Figure 13.** Evaluation of sample predictions using different `maxperimage` settings for *LineFormer* with the peak-aware *Custom* metric.

**Table 4.** Accuracy (Custom peak-aware score) of sample predictions with different settings of `maxperimage` *LineFormer* hyperparameter, sorted by mean average accuracy.

| maxperimage | selector | count | mean | std | min | 25% | 50% | 75% | max |
|---|---|---|---|---|---|---|---|---|---|
| **100** | **by confidence** | 180 | 0.884 | 0.233 | -0.137 | 0.957 | 0.978 | 0.988 | 0.999 |
| **1** | **n/a** | 180 | 0.951 | 0.092 | 0.167 | 0.955 | 0.977 | 0.988 | 0.999 |
| **2** | **by confidence** | 180 | 0.899 | 0.210 | -0.125 | 0.957 | 0.978 | 0.988 | 0.999 |
| **3** | **by confidence** | 180 | 0.892 | 0.221 | -0.125 | 0.956 | 0.978 | 0.988 | 0.999 |
| **100** | **by visual inspection** | 180 | 0.968 | 0.055 | 0.474 | 0.967 | 0.980 | 0.990 | 0.999 |
| **100** | **with post correction** | 180 | 0.979 | 0.021 | 0.827 | 0.972 | 0.983 | 0.993 | 1.000 |

In summary, the experimental pilot study revealed that high-resolution scans of full historical pages are unsuitable for direct use with *LineFormer*. Instead, a tailored pre-processing approach combining monthly cropping with x-axis stretching and introducing human visual inspection for candidate selection as well as manual post corrections, was required to unlock highly usable predictions from the model. These insights were instrumental in shaping our production workflow.





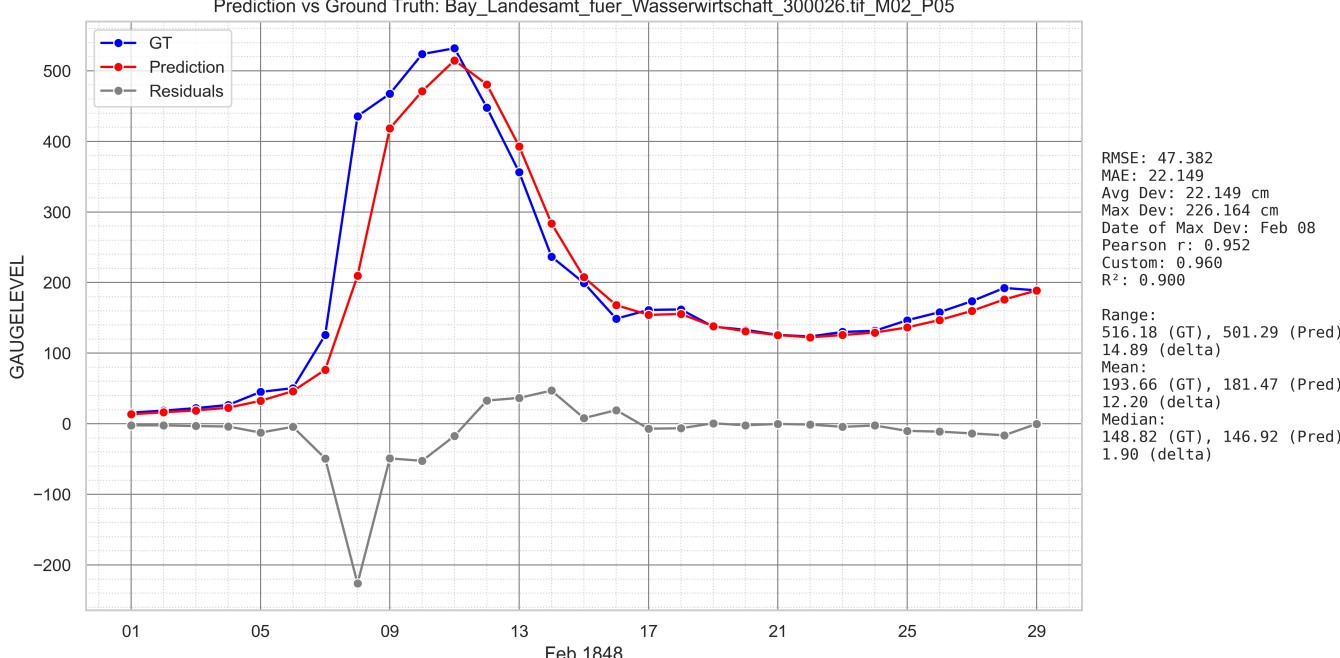

**Figure 14.** Average prediction (0.960) in the sample set according Custom peak-aware metric (*Passau, Feb. 1848*). It shows a time bias of one day delay and consequently, due to the steep incline, a maximum deviation of 2.26 metres on that day (Feb 8). Despite the delay, the overall shape and the peak is quite accurate, hence a high Pearson r of 0.952.

### 4.5.6 Evaluation

The evaluation for the complete sample set and the finally chosen setup with $maxperimage = 100$, candidate selection by human visual inspection and optional post correction results were already given (Tab. 4). Fig. 14 gives an impression of what

an average prediction of 0.968 accuracy can mean; it shows the gauge month prediction closest to the mean average according to custom peak-aware score and features a time bias.

Fig. 15 shows the worst prediction of the sample with a custom peak-aware accuracy of 0.359 and a negative $r = -0.239$. At the current stage of research, no setting could be developed to improve this prediction significantly; it was hence flagged 'M' for major manual correction required.

### 4.5.7 Systematic error analysis

We examined recurring failure modes across gauges, months, and candidate settings to identify systematic (i.e. non-random) error sources (cf. Fig. 16). Below we summarise the main patterns, in addition to those described in Section 3.1:


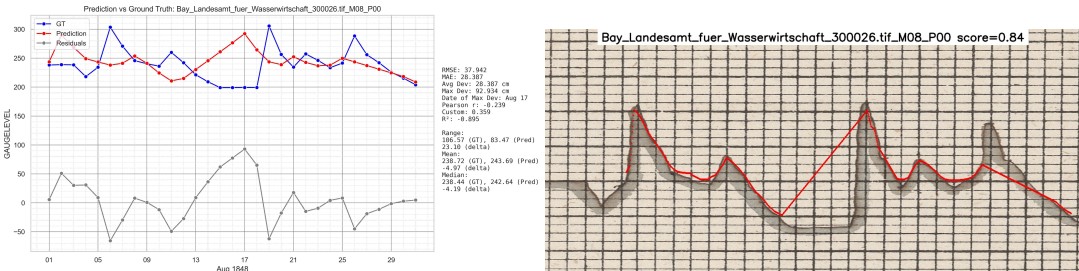

**Figure 15.** Worst prediction in the sample set according Custom peak-aware metric (0.359, $Pearson\ r = -0.239$). The right part of the image shows the results directly from the *LineFormer* prediction. While there are parts of the charts predicted nicely, there are three issues: (1) the incompleteness of the predicitons, leaving out the first four days; (2) not recognizing correctly the transition between 13 and 18 Apr and at the end of the months; and (3) the time shift due to the missing data during curve-to-series transformation (*Passau, Aug. 1848*). Image source: BayHStA BLW 30.

- **Grid-parallel strokes and steep segments.** When the hydrograph runs almost parallel to the printed grid or coincides with it (especially near vertical segments at floods), local contrast with gridlines decreases and LineFormer may fragment or hop tracks or detection confuses ink with print.

- **Extreme inclines.** Near-vertical segment runs parallel to the grid; day bins cut through a very steep slope causing peak assigned to two consecutive days.

- **Shadow hiding data.** Stroke shadowing below the curve plus extreme slope leads to under-prediction or missing the true apex.

- **Shadow pulls prediction below line.** Local paper darkening/ink bleed forms a halo under the stroke; predicted track sits a few pixels below the true curve.

- **Undecidable data source.** The way in which the chart is drawn does not allow a precise dating.

- **Missing or superfluous data.** The drawn chart is either prolonged over the first or last day of month, or days are missing, both leading to inaccuracy and a time bias.

Systematic errors primarily affect extremes (peak timing/height) and month-transition continuity. The adopted mitigations improve robustness without increasing manual load substantially; remaining edge cases motivate targeted fine-tuning on historical artefacts, automated panel/axis detection with confidence-gated fallbacks, and tighter dewarping–calibration integration in future work.

### 4.5.8 Benchmarks

Tab. 5 shows a benchmarking of different approaches evaluated in order to discuss effect (accuracy) and efficiency (time cost). As discussed above (Sec. 4.4.3), the two manual approaches have a strong agreement (0.996). With the manual pixel-based



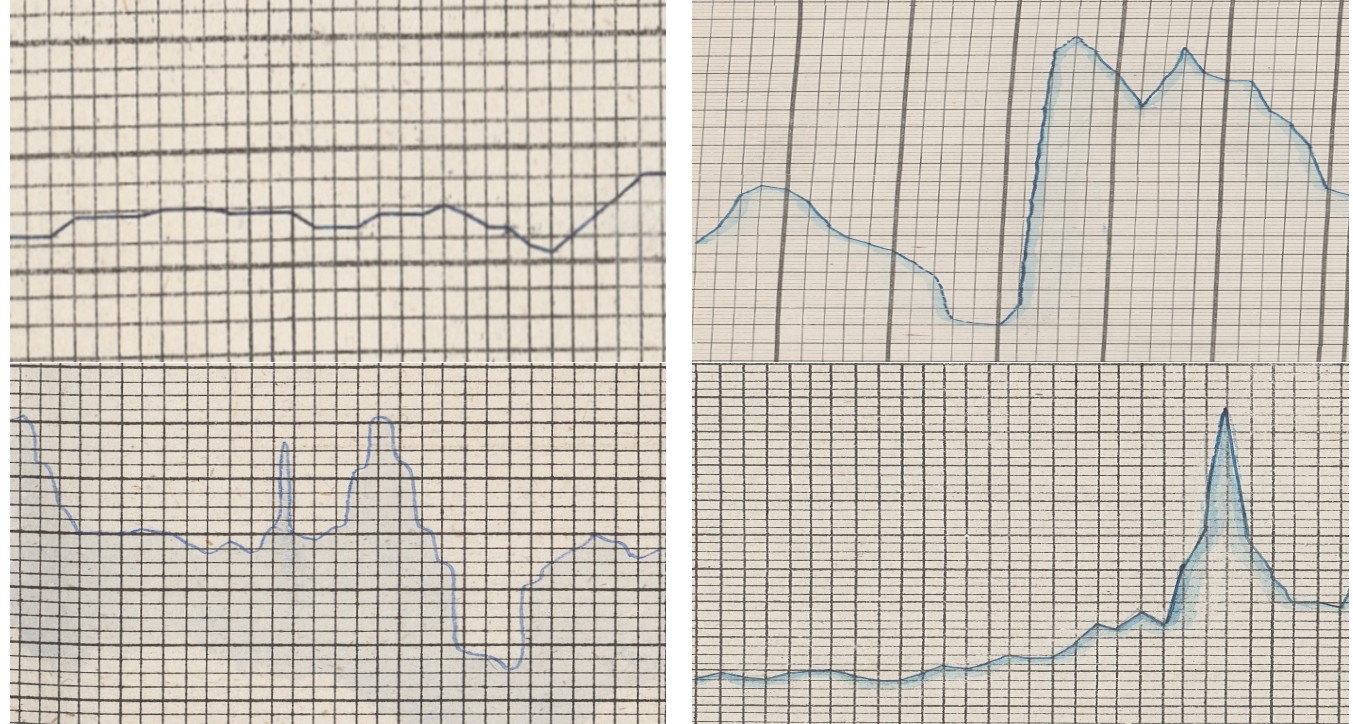

**Figure 16.** Samples illustrating different systematic sources of errors. Top left: Horizontal overlap of grid and graph (*Neu-Ulm, Oct. 1865*); top right: near vertical stroke in mid of month (*Passau, Jan. 1878*); bottom left: undecidable day of peak around 14 July (*Passau, July 1845*); bottom right: month of April displaying 31 instead of 28 values (*Neu-Ulm, Apr. 1871*). Image sources: BayHStA BLW 21, 30.

annotation serving as ground truth, either manual approach makes the gold standard for data rescuing historical gauge level charts in terms of accuracy.

In terms of efficiency, the semi-automated approaches decrease costs in form of manual labour by factors 9.62 to 6.10 respectively. For the production process, we chose to include manual post correction with is an overall trade-off of manual labour saving of 1,505 sec. per gaugeyear and an accuracy drop to 0.977. We also assess that refraining from manual post correction (accuracy drop to 0.964 at 1,577 sec. savings per gauge year) is also a viable way.

## 4.6 Validation

For validation, we created another test set by randomly selecting five gauge years that were not part of the tuning sample: Neu-
Ulm 1833 (ID 210011), Vilshofen 1849 (ID 290027), Passau 1845 (ID 300023), 1861 (ID 300042), and 1871 (ID 300052). Running the complete pipeline with the settings as described above but without manual post correction, we achieved a mean accuracy of 0.954 on Custom peak-aware score against a manual pixel-based annotation. This is within the range of the sample set which yielded 0.870 and confirms the set-up.





**Table 5.** Efficiency / effectiveness comparison of different evaluated approaches. Compare with Fig. 17.

| approach | N | mean accuracy | mean time | time cost ratio |
|---|---|---|---|---|
| **manual keying** | 48 | 0.996 | 8,400 | 4.67 |
| **manual pixel-based annotation (GT)** | 180 | 1.000 | 1,800 | 1.00 |
| **by confidence** | 180 | 0.884 | 187 | 0.104 |
| **by visual inspection** | 180 | 0.968 | 223 | 0.124 |
| **with post correction** | 180 | 0.979 | 295 | 0.164 |

Manual pixel-based annotation serves as benchmark. Efficiency measured in seconds per gauge year; time cost ratio = (seconds per gauge-year) / 1,800 (GT baseline); effectiveness (accuracy) in Custom peak-aware score. Manual processes include double-keying but no post-correction. Other estimates as described above: baseline annotation: 27 sec./gaugeyear, visual inspection: 36 sec./gaugeyear, post correction: 60 sec./gaugemonth with 10% required.

### 4.7 Final workflow

Taking these consideration and pre-experiments into account, we introduce in the following our pipeline *Historical Water Level Reconstruction (HWLR)* as summarised in Fig. 17.

For each gauge year the pipeline processes (a) a high-resolution annual chart scan; (b) station metadata (identifier, year, unit system), including any known unit transitions; (c) twelve monthly bounding boxes (annotated or detected); (d) two vertical anchors per month (e.g. lowest and highest labelled grid marks with pixel and gauge level values); and (e) a horizontal baseline 610 to detect possible image wraping.

For each gauge month the pipeline produces: a set of (x,y)-pixel coordinates with confidence score as the presumably best prediction of the actual drawn line chart (usally around 2,000 per gauge month) and a daily time series in physical units (millimetres) matched from interpolated (x,y)-pixel coordinates and based on the individual gauge zero mark. Pixel coordinates are measured on the full-resolution scan with origin at the top-left; y increases downward using YOLO-style percentages of 615 the total image size.

### 4.8 Publication

After a final automatic sanity check for completeness, the data was compiled into a dataset in CSV-format. Each row of the dataset represents one data point, i.e. one observation at a particular gauge at a particular time. The dataset is structured in the following columns:

– **obs_id:** unique identifier for this data point

– **gauge_id:** gauge of this data point

– **date:** date of observation (`YYYY-MM-DD`)

– **waterlevel_mm:** observation in millimetres as reconstructed





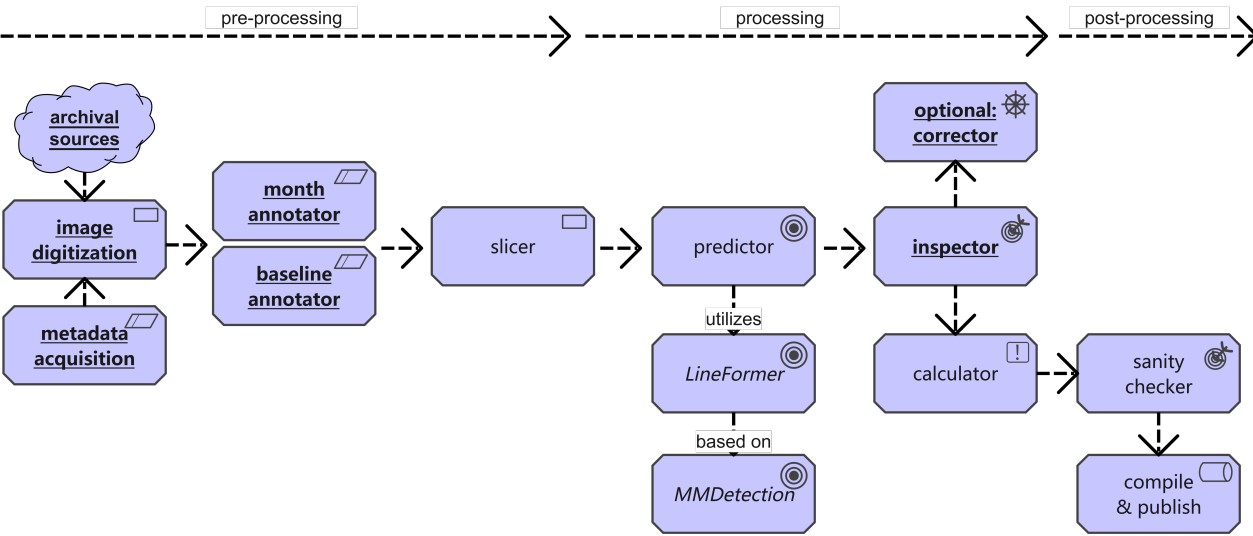

**Figure 17.** Final *Historical Water Level Reconstruction (HWLR)* production workflow. Underlined tasks require human support.

- **confidence:** confidence score [0,1] of automated prediction

– **coverage:** percentage [0,1] of the area on the x-axis that is covered by data points; ideally close to 1

- **method_id:** method used for this data point

- **data_reconstruction:** date of reconstruction (`YYYY-MM-DD`)

- **responsible_id:** person or institution responsible for this data point

- **source_id:** identifier refering to original archival source (image-based)

– **notes:** annotations provided by reconstruction method

- **pixel_x:** normalised [0,1] x-pixel-coordinates of this data point on the prediction image (YOLO-format)

- **pixel_y:** normalised [0,1] y-pixel-coordinates of this data point on the prediction image (YOLO-format)

This dataset is linked to a second dataset documenting details about the gauges as archival sources as follows. Each row of the dataset represents one archival transmission of gauge records for one gauge. The dataset is structured in the following

columns:

- **gauge_id:** unique identifier in the format CC_FF_RRR_n where CC is a country code (e.g. DE=Germany), FF a federal state or regional code (e.g. BY=Bavaria), RRR a river code (e.g. DAN for Danube), and n a running number. For the





material from the Bavarian State Archives, the running number refers to the archival identifier (call number) within its larger collection.

– **gauge_name:** name of the gauge as stated in the source

   – **river_name:** name of the river in English or vernacular language

   – **longitude, latitude:** geographic data of the gauge location

   – **osm_id:** OpenStreetMap identifier of the gauge location

   – **altitude:** altitude of the gauge location in metres above sea level

– **reporting period:** time span covered by this archival record

The data for this particular study has been published in an open online repository under the Creative Commons CC BY 4.0 International licence (Rehbein, 2025).

### 4.9 Sample Study

To illustrate potential uses of the dataset, we analyse flood and arid (low-water) periods using the reconstructed *stage* (water-
level) series. Thresholds are computed *per calendar month* to respect intra-annual variability.

We define flood days as those on which daily stage exceeds the monthly 95th non-exceedance percentile, with a minimum duration of at least three consecutive days to avoid one-day spikes. This follows ecological-hydrology practice that emphasises frequency, duration, timing, and rates of change of high stages.

Arid (low-water) periods are defined analogously using the monthly 5th non-exceedance percentile of daily stage, again
requiring at least three consecutive days. This mirrors standard low-flow statistics (often discussed via "Q95" low-flow metrics in European practice) while preserving ecological meaning by accounting for seasonal minima.

Without further interpretation, Figures 18 and 19 report flood and arid periods for the Danube gauges Neu-Ulm, Vilshofen, and Passau from 1826 to 1894.

### 5 Conclusions

This study demonstrates that nineteenth-century, hand-drawn river gauge charts can be systematically transformed into machine-readable daily water-level series with transparent uncertainties and page-level provenance. We present a pragmatic, semi-automated pipeline (HWLR) that combines light, grid-aware pre-processing, transformer-based line extraction, and targeted human oversight. On our Bavarian Danube case study (1826–1894; three gauges), the workflow achieves high accuracy at series level while reducing manual effort by an order of magnitude relative to fully manual digitisation. The resulting datasets,
code pointers, and processing artefacts are openly released to support audit, reuse, and extension.




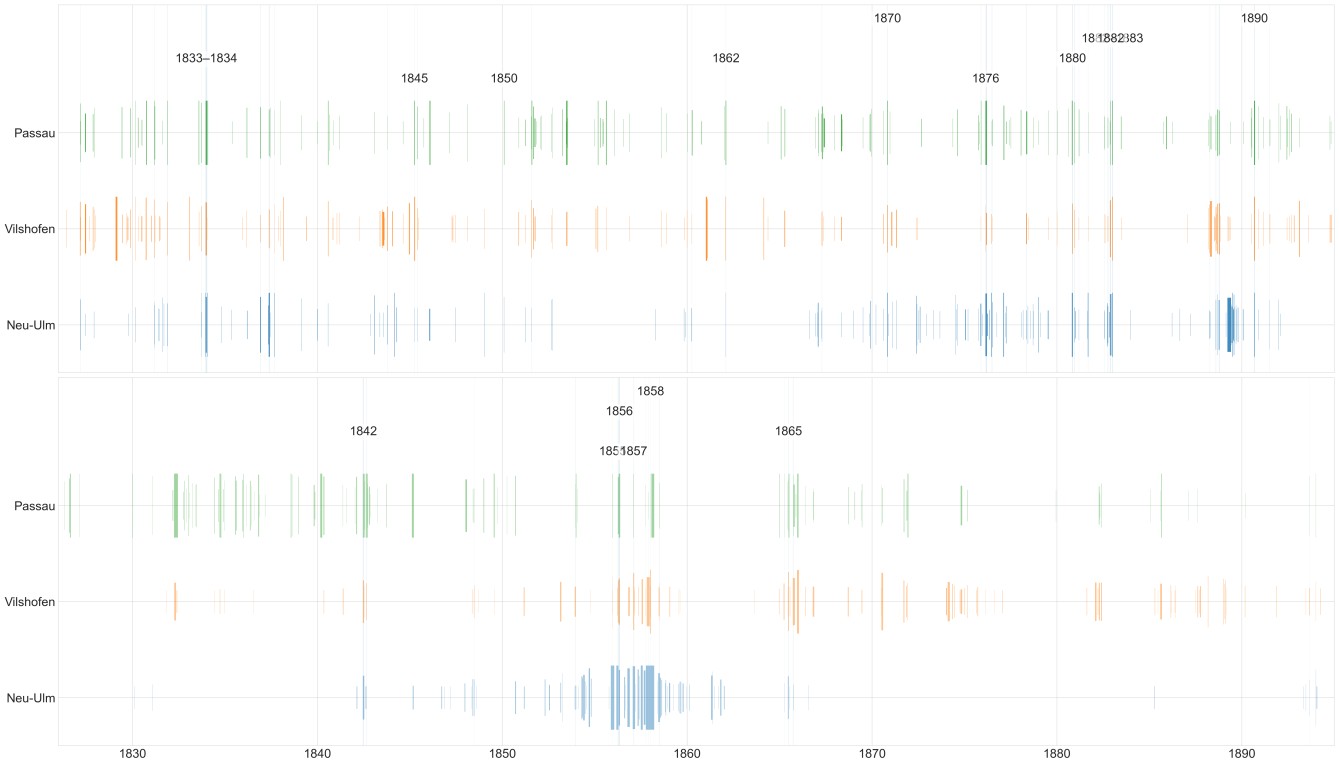

**Figure 18.** Flood (top) and arid periods along the Danube gauges Neu-Ulm (DE_BY_DAN_21), Vilshofen (DE_BY_DAN_29), and Passau (DE_BY_DAN_30) for the years 1826–1894.

Beyond the immediate case, the approach provides a replicable methodological and practical template for mobilising large analogue hydrometric holdings. By tying reconstruction explicitly to gauge datums and documenting unit transitions, the series produced here can underpin long-term hydrological analyses, before/after assessments of nineteenth-century river works, and multi-decadal context for extremes. The open release also invites community validation and iterative improvement.

While effective on heterogeneous archival material, several limitations remain. First, specific drawing styles and shadowed strokes occasionally degrade line following, indicating value in fine-tuning or training model variants on curated historical samples. Secondly, two manual steps—monthly panel detection and y-axis anchor extraction—are currently the main bottlenecks; robust automation of both would further reduce effort and variance. Thirdly, automatic candidate selection still underperforms rapid visual inspection in edge cases; incorporating coverage-aware scoring, better uncertainty estimates, and modest domain

adaptation should narrow this gap. Finally, mild residual page warp can bias day assignment where grid geometry departs from ideal; tighter integration of document dewarping with coordinate recovery is a promising avenue.

Scaling to the full archival series is a priority; the present release should be seen as a foundation rather than an endpoint. Reconstruction at scale requires not only techniques but also sustained curation. We align this work with ongoing international data-rescue efforts and propose maintaining the Bavarian Danube series as a "living data" resource with versioned updates,





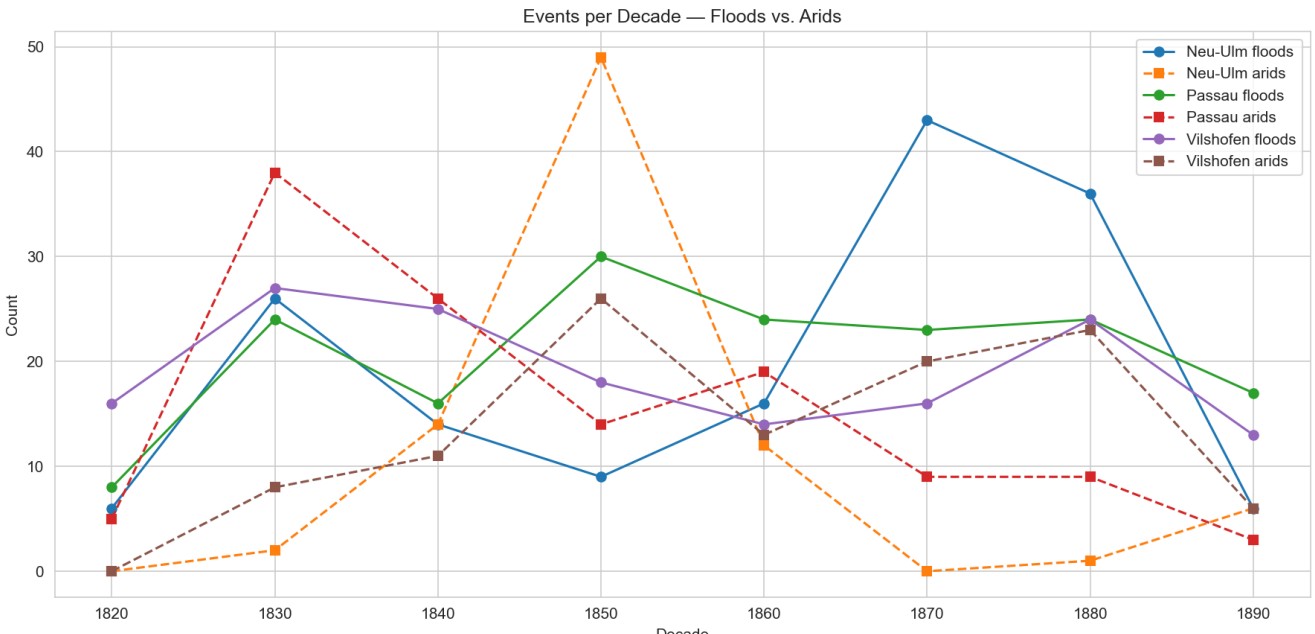

**Figure 19.** Count of flood and arid events along the Danube gauges Neu-Ulm (DE_BY_DAN_21), Vilshofen (DE_BY_DAN_29), and Passau (DE_BY_DAN_30) for the years 1826–1894.

expanded station coverage, and clear provenance metadata. The current dataset is openly available, providing an immediate basis for hydrological, geomorphological, and ecological studies.

## 6  Code and data availability

The data for this particular study has been published in an open online repository under the Creative Commons CC BY 4.0 International licence DOI:10.5281/zenodo.17296750 (Rehbein, 2025). *LineFormer* is available at https://github.com/TheJaeLal/
LineFormer (Lal et al., 2023).

*Author contributions.*  MR is the sole author, developer, and data producer.

*Competing interests.*  The author declares no competing interest.





*Acknowledgements.* This work benefited greatly from the support and access provided by the Generaldirektion der Staatlichen Archive Bayerns and the Bayerisches Hauptstaatsarchiv München. I am grateful to the Computational Historical Ecology research group at the
University of Passau for many valuable discussions. The technical workflow builds on *LineFormer* and its foundations; I thank the developers and maintainers for their outstanding contributions. ChatGPT 5 was used as a coding assistant. DeepL and ChatGPT5 supported translations and English stylistics.





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
