# Peer review of "Reconstructing Nineteenth-Century River Water Levels with Transformer-Based Computer Vision"

_Earth System Science Data, 2025_

## Referee Comment (RC1)

This paper presents an efficient methodology for rescuing daily hydrometric data that are stored only in hand-made graphs of time series. It is very interesting because data rescue (DARE) efforts are usually more focused on meteorological data than on hydrometric records, and this is not because the latter are less important than the former. Also the proposed methodology is useful as it can be applied to data from many disciplines, not only from hydrology. Below I present some major comments and questions followed by few minor comments.

Major comments and questions:

1. Could you provide technical information about the image quality required for the proposed DARE methodology? Is there a minimum resolution (DPI) required? I only found some generalities on this matter. For instance, in lines 238-239 you can read: "Typical workflows comprise high-resolution scanning".

2. In connection to previous comment, I wonder if you worked (or plan to work) with photographs rather than scans. I'm familiar with rescuing meteorological data where it is advised to photograph large amounts of documents because this imaging procedure is many times faster than scanning (see Section 2.4.2. of Wilkinson, 2019). You developed an efficient way of obtaining machine-readable data from graphs and it would also make sense to accompany it with an efficient imaging strategy. Although this imaging issue does not matter much if al the Danube water level charts were already scanned with high resolution.

3. In the second paragraph of Section 3.1 you can also mention the methods developed to extract subdaily data from strip chats of meteorological instruments such as thermographs and barographs (e.g., Sušin and Peer, 2018).

4. In section 3.2.1 entitled "Americas" you can cite recent works that rescued long Paraná River hydrometric records, which start in 1875. Indeed Antico et al. (2018) manually digitized daily water levels from a hand-drawn chart similar to the one shown in the upper left panel of Fig. 5 of the revised manuscript. More recently Antico et al. (2020) found the tabulated version of these data and compared tabulated values with those digitized from the chart.

5. Did you consider using documentary sources (e.g., newspapers) or metadata provided by the charts to correct time misalignments of positive and negative peaks (floods and drought)? This could be a useful correction, as knowing the exact date of these peaks is important for many studies.

6. Similarly, documentary sources may inform the exact river levels attained during these peaks. That is, these sources may serve to correct these levels.

Minor comments:

1. In the title, it could help to replace "Nineteenth-Century River Water Levels" by "Nineteenth-Century Danube River Water Levels"

2. Line 17: replace "(see Fig. 1) for anexample)" by "(see Fig. 1 for anexample)".

3. Lines 233-234: replace "software tools such as TIITBA Corona-Fernandez and Santoyo (2023)" by "software tools such as TIITBA (Corona-Fernandez and Santoyo, 2023)"

References:

Antico, A., Aguiar, R. O., & Amsler, M. L. (2018). Hydrometric data rescue in the Paraná River Basin. Water Resources Research, 54(2), 1368-1381. https://doi.org/10.1002/2017WR020897

Antico, A., Mendizabal, S., Ferreira, L. J., Aguiar, R. O., & Amsler, M. L. (2020). Addendum to "Hydrometric Data Rescue in the Paraná River Basin" by Andrés Antico, Ricardo O. Aguiar, and Mario L. Amsler. Water Resources Research, 56(2), e2019WR026654. https://doi.org/10.1029/2019WR026654

Wilkinson, C., Brönnimann, S., Jourdain, S., Roucaute, E., Crouthamel, R., Brohan, P., ... & Gilabert, A. (2019). Best Practice Guidelines for Climate Data Rescue v1, of the Copernicus Climate Change Service Data Rescue Service. https://doi.org/10.24381/x9rn-mp92

Sušin, N., & Peer, P. (2018). Open-source tool for interactive digitisation of pluviograph strip charts. Weather, 73(7), 222-226. https://doi.org/10.1002/wea.3001